# Structural plasticity of bacterial ESCRT-III protein PspA in higher-order assemblies

Benedikt Junglas [1,8], Esther Hudina [1,2,8], Philipp Schönnenbeck[1,2], Ilona Ritter[1], Anja Heddier[1,2], Beatrix Santiago-Schübel [3], Pitter F. Huesgen [3,4,5], Dirk Schneider [6,7] & Carsten Sachse [1,2] ✉

Eukaryotic members of the endosome sorting complex required for transport-III (ESCRT-III) family have been shown to form diverse higher-order assemblies. The bacterial phage shock protein A (PspA) has been identified as a member of the ESCRT-III superfamily, and PspA homo-oligomerizes to form rod-shaped assemblies. As observed for eukaryotic ESCRT-III, PspA forms tubular assemblies of varying diameters. Using electron cryo-electron microscopy, we determined 61 *Synechocystis* PspA structures and observed in molecular detail how the structural plasticity of PspA rods is mediated by conformational changes at three hinge regions in the monomer and by the fixed and changing molecular contacts between protomers. Moreover, we reduced and increased the structural plasticity of PspA rods by removing the loop connecting helices α3/α4 and the addition of nucleotides, respectively. Based on our analysis of PspA-mediated membrane remodeling, we suggest that the observed mode of structural plasticity is a prerequisite for the biological function of ESCRT-III members.

In eukaryotes, ESCRT proteins form a multi-subunit machinery that performs a topologically unique membrane budding reaction away from the cytoplasm[1]. The ESCRT system assumes critical roles in many cellular processes, including nuclear envelope sealing[2], plasma membrane repair, lysosomal protein degradation[3], retroviral budding and the multivesicular body pathway[1]. The biological activity of the conserved group of ESCRT-III proteins appears to be linked to the formation of homopolymeric and/or heteropolymeric structures, such as sheets, strings, rings, filaments, tubules, domes, coils and spirals[4–7]. It was recently shown that the bacterial proteins PspA and the vesicle-inducing protein in plastids 1 (Vipp1, also known as the inner membrane-associated protein of 30 kDa, IM30) are structurally similar to ESCRT-III proteins found in eukaryotes and archaea[8–10] and indeed they have a common ancestor, thereby extending the ESCRT-III superfamily to the bacterial domain[10,11].

The 25.3 kDa PspA is the main effector of the *Escherichia coli* Psp system that protects and maintains the bacterial inner membrane[12,13]. In bacteria, PspA is present both in a soluble form in the cytoplasm and bound to negatively charged lipids of the inner membrane[14,15]. Thus, PspA can counteract membrane stress and block the leakage of protons through damaged membranes[13,16,17]. Another well-studied example of a PspA-like protein that has an essential role in membrane protection but also in membrane remodeling (that is, of the thylakoid membrane) is Vipp1. Vipp1 is capable of membrane fusion as a means of active membrane repair as well as passive membrane protection of a protein carpet forming on the membrane[18–20]. PspA and Vipp1 both consist solely of α-helices connected by short loops (Vipp1, seven α-helices; PspA, six α-helices) with a four-helix core structure, in which α1 and α2/3 form a coiled-coil hairpin connected by a short loop[8–10].

[1]Ernst-Ruska Centre for Microscopy and Spectroscopy with Electrons, ER-C-3/Structural Biology, Forschungszentrum Jülich, Jülich, Germany. [2]Department of Biology, Heinrich Heine University, Düsseldorf, Germany. [3]Zentralinstitut für Engineering, Elektronik und Analytik (ZEA-3), Forschungszentrum Jülich, Jülich, Germany. [4]Cluster of Excellence on Aging-related Disorders (CECAD), Medical Faculty and University Hospital, University of Cologne, Cologne, Germany. [5]Institute of Biochemistry, Faculty of Mathematics and Natural Sciences, University of Cologne, Cologne, Germany. [6]Department of Chemistry, Biochemistry, Johannes Gutenberg University Mainz, Mainz, Germany. [7]Institute of Molecular Physiology, Johannes Gutenberg University Mainz, Mainz, Germany. [8]These authors contributed equally: Benedikt Junglas, Esther Hudina. ✉e-mail: c.sachse@fz-juelich.de

A distinctive feature of PspA and Vipp1 is their ability to form MDa-sized ring-shaped or rod-shaped homo-oligomers, as observed also with eukaryotic ESCRT-IIIs[8–10,21,22]. PspA rods are helical assemblies with the monomers arranged by a brick-like stacking, while the four-helix core of α1–α4 forms the wall of the tubular PspA rods, with α5 pointing to the outside[9]. The rod structure of PspA of the cyanobacterium *Synechocystis* sp. PCC 6803 (hereafter PspA) with a diameter of about 200 Å was recently determined by cryo-electron microscopy (cryo-EM), and PspA was shown to form diverse rods having variable diameters and lengths of up to several hundred nanometers[9].

Eukaryotic ESCRT-III monomers have been observed in different conformational states within heteropolymeric complexes of different ESCRT-III isoforms[23,24]. However, the structural flexibility of ESCRT-III proteins in homopolymers and the enabling structural mechanisms have not been studied in molecular detail. Here, we solved the structures of 61 PspA rods using cryo-EM and observed a wide range of finely sampled assemblies with different rod diameters displaying notable structural plasticity. We show that nucleotide addition affects the distribution of rod diameters. In the absence of nucleotides, PspA rods have narrow diameters. In the presence of ADP, PspA rods shift to wider diameters, and in the presence of ATP, PspA rods have the widest diameters. We show that the addition of ATP enhances the membrane remodeling activity of PspA, indicating that structural plasticity may be critical for the ability of PspA to engulf and remodel membranes. Comparing 11 PspA structures with increasing diameters, we observed that the structural plasticity of PspA rods requires conformational changes at three defined hinge regions. In these structures, the length of α3 increases as the loop connecting α3 and α4 shortens within larger diameter rods. Upon deletion of the loop connecting α3 and α4, we observed a significant reduction of the structural plasticity and no effect on the diameter distribution upon addition of ATP. In summary, PspA monomers display a remarkable structural plasticity allowing the assembly of multiple stable rod structures with varying diameters.

## Results

### PspA forms helical rod assemblies with different diameters

To investigate the structural heterogeneity observed in previous PspA analyses[9], we prepared PspA rods by refolding PspA purified in denaturing conditions and observed the formation of rods with variable diameters ranging from 180 Å to 250 Å with a maximum at 215 Å (Fig. 1a). After multiple rounds of classification and symmetry analysis of helical segments (Fig. 1b and Extended Data Fig. 1a), we determined the cryo-EM structures of rods that were 180, 200, 215, 235 and 250 Å in diameter at between 4.2 Å and 6.2 Å resolution. The 200 Å rod structure determined at 4.2 Å resolution is highly similar to the previously published structure of PspA rods (PDB 7ABK, EMDB-11698), whereas increasing diameters showed lower resolution (Fig. 1c). The cross-sectional top and side views of the cryo-EM maps show features expected for PspA rods (spikes formed by the α1/2 + 3 hairpin, Christmas-tree pattern in the cross-sectional side view). The continuous series of PspA cryo-EM structures confirms that PspA can form a range of different stable assemblies, indicating structural plasticity.

### Structural plasticity of PspA can be enhanced by nucleotides

Vipp1 was shown to form tapered ring structures with different conformations and diameters inside a single determined assembly[8]. As nucleotides were identified in the narrowest ring, it had been suggested that ADP and/or ATP binding affected the diameter of the assembly. Inspired by this observation, we added ADP during filament formation and measured a diameter distribution broadened to a range from 180 Å to >400 Å with a maximum at 250 Å, whereas the fraction of rods having a diameter of >280 Å did not exceed 5% (Fig. 2a). Based on this dataset, we determined a total of 11 additional cryo-EM structures (at 4.3–7.6 Å global resolution) for the respective diameters. A total of five (180, 200, 215, 235 and 250 Å) out of the 11 structures completely overlapped with the structures solved above in the absence of nucleotides. The remaining six rod structures showed increased diameters while rods with diameters of >320 Å contained additional cylindrical density in the lumen. The corresponding micrographs suggested that 'super rods' were formed by wide rods engulfing narrower PspA rods of different helical symmetry and appeared as smooth cylinders after the imposition of helical symmetry (Extended Data Fig. 1b). Finally, we also incubated PspA with ATP during filament formation and determined a total of 11 additional cryo-EM structures at resolutions ranging from 3.8 Å to 6.9 Å (Table 1 and Fig. 2b). Notably, for rod diameters of 180, 200, 215 and 235 Å, we identified density in the putative nucleotide binding site close to the loop connecting helices α3 and α4 (amino acids 155–156), albeit weaker than the surrounding protein density, presumably because of incomplete binding (Extended Data Fig. 1c). Closer inspection of the binding pocket and comparison with the PspA apo model revealed that the nucleotide competes with the α3/α4 loop, which results in an alternative conformation and reduced mobility of the α3/α4 loop, associated with lower atomic temperature factors, when the nucleotide is present (Extended Data Fig. 1d). Although some of the rod structures were already identified in the presence and absence of ADP, the structures of three additional diameters at 290, 305 and 320 Å could be resolved solely when ATP was present. As they represent a large range of structures observed under the same experimental conditions at a suitable resolution (Extended Data Fig. 2a–f), we built the corresponding 11 atomic models (Fig. 2c). In total, we determined 27 structures of PspA assemblies using cryo-EM in three conditions of nucleotide absence, ADP and ATP, with 14 unique structures differing in their respective diameters, thus enhancing the observed structural plasticity of PspA assemblies.

### Hinge regions serve as dials for PspA's structural plasticity

To analyze the detailed molecular mechanism underlying the structural plasticity of PspA, we scrutinized the 11 built atomic models. The monomer structures were overall similar or nearly identical to the published PspA structure (PDB 7ABK) in the case of the 200 Å diameter rods, except for deviations in the loop connecting α3 and α4 (Fig. 3a and Extended Data Fig. 2g). After superimposition of PspA monomer structures from different diameters, we found that α1–α5 were almost unchanged despite changes in their relative orientation (Fig. 3b and Extended Data Fig. 3a). Additionally, we observed that α3 and α4 become extended with increasing rod diameters, resulting in a shortening and ultimate disappearance of the loop between these helices in the widest rods (320 Å and 365 Å diameters). Thus, α3 and α4 eventually fused to form a single α-helix. The length of α5 and the loop connecting helices α4 and α5 remained unaffected. The measured atomic distances from Cα-G82 to Cα-P187 increased for PspA rods with smaller diameters (180–270 Å), whereas they decreased for diameters larger than 270 Å (Fig. 3c). Thus, PspA adopted a bow-like shape under tension for the smaller diameters, up to an almost linear shape at around 270 Å diameter and a slightly inverted bow curvature for the largest diameters. The structural changes were realized by three hinge regions in the monomer: the first hinge was positioned at the transition between α2 and α3 (amino acids 128–133), the second hinge was located in the loop region between α3 and α4, and the third hinge is in the loop region between α4 and α5. The largest angular changes came from the second hinge (−45° to 20°), whereas the angular changes of the other two hinges were shallower (Fig. 3d).

To examine the contacts between protomers within rod assemblies of different diameters, we next determined the Cα-distance changes over the different diameters between evolutionarily conserved residues of neighboring PspA monomers with respect to the smallest 180 Å diameter rods and computed the s.d. as a measure of change (Fig. 3e and Extended Data Fig. 3b,c). The observed s.d. values fell into two major groups: contacts with distance changes smaller than 1 Å and larger than 1 Å, which we assumed to keep fixed contacts or switch contacts

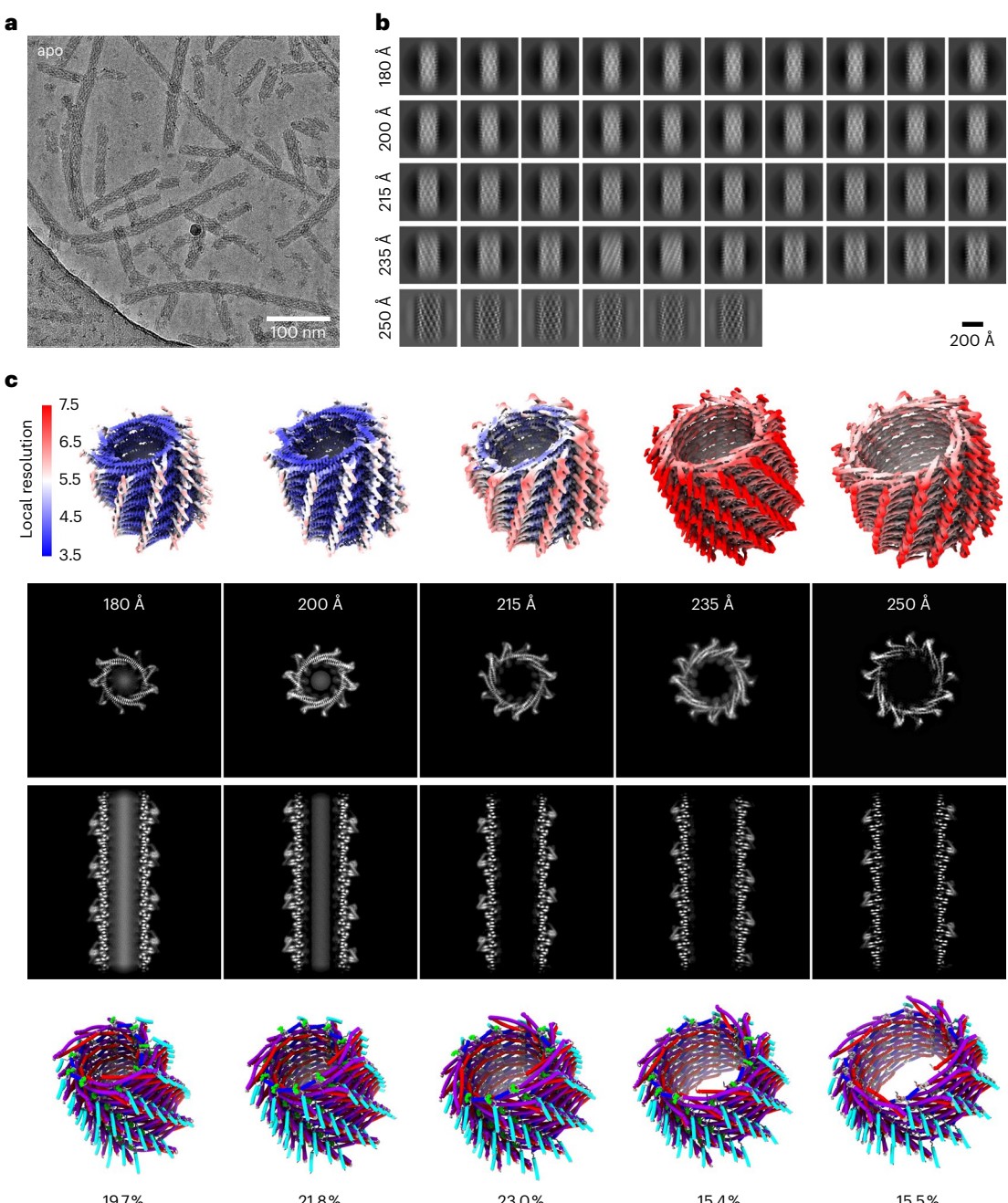

**Fig. 1 | PspA self-assembles into a series of different rods with diameters from 180 to 250 Å. a**, Selected cryo-EM micrograph from 1,818 recorded movies of PspA revealing different rods with respect to the length and width of the observed assemblies. **b**, Image classification of helical segments into five different groups with diameters of 180, 200, 215, 235 and 250 Å, respectively. **c**, Five corresponding three-dimensional cryo-EM structures colored with local resolution estimates (global resolution of 4.3, 4.2, 4.6, 5.7 and 6.2 Å, respectively). Cross-sectional top and side view (top) of the five structures with increasing diameters including the atomic models (bottom) with the respective relative abundance, with helices colored as follows: α1, red; α2 + 3, violet; α4, blue; α5, cyan.

with increasing diameters, respectively. The fixed intermolecular contacts consisted of electrostatic and hydrophobic interactions. The corresponding residues are clustered in two regions at the hairpin α1/α2–α5 interface (W71, K74, R88, L91, F196 and M212) and the α2 + 3–α1/α2 hairpin interface (R44, K55, L125, K128, L138 and R142). Additional fixed residues can be found at the start and end of α4 (F168 and E179) (Fig. 3f). Fixed contacts probably serve as anchor points that are critical for keeping the polymer assembled during structural rearrangements resulting in diameter transitions. In addition to these fixed contacts, we identified several evolutionarily conserved contacts that have large Cα-distance changes and probably switch their interacting residues

when comparing different diameters (E126, R170 and K174). Together with a hydrophobic groove in the hairpin between α1 and α2 formed by mostly non-conserved residues (Fig. 3g), they provide the flexibility to allow subunits to slide relative to each other accompanied by the described conformational change of the monomers. Furthermore, the helical rise of the rods decreases with increasing diameters, resulting in a linear mass-per-length increase (Extended Data Fig. 3d–f). In conclusion, the structural plasticity of PspA rods is accomplished by a combination of evolutionarily conserved fixed and switching residue interactions and of non-conserved hydrophobic groove interactions between neighboring PspA molecules.

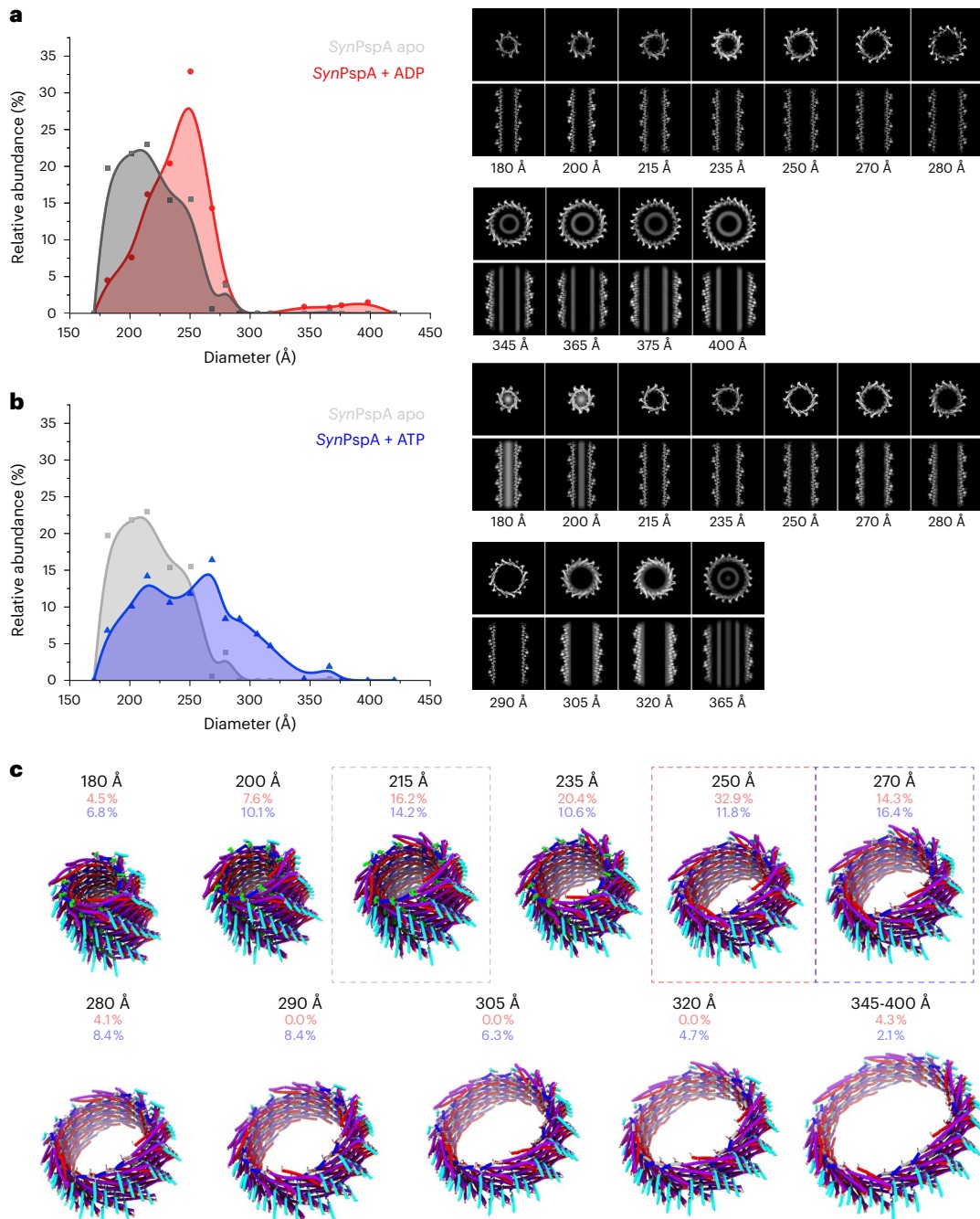

**Fig. 2 | The structural plasticity of PspA rods increases upon incubation with ADP and ATP. a**, Histogram of rod diameters and the relative abundance observed upon addition of ADP (red) during refolding in comparison with PspA apo (gray). In the presence of ADP, the distribution maximum shifts to 250 Å, and diameters up to 400 Å could be observed. The associated cryo-EM structures are shown in the cross-sectional top and side views. **b**, Histogram of rod diameters

upon addition of ATP (blue) during refolding in comparison with PspA apo (gray). In the presence of ATP, the rod population shifts further towards larger diameters in both cases. **c**, Corresponding three-dimensional cryo-EM atomic models. Dashed-line frames highlight the most frequent structure for apo (215 Å, gray), ADP (250 Å, red) and ATP (270 Å, blue) (α1, red; α2 + 3, violet; α4 blue, α5 cyan).

## PspA rods bind and hydrolyze ATP

Based on our observations and the recently identified NTPase activity of Vipp1 (refs. [8],[25]–[27]), we investigated whether PspA has a similar ATPase activity. In our structures, we could not identify a canonical P-loop or Walker A motif but previous Vipp1 mutations in conserved residues R44, E126 and E179 had significantly decreased the ATPase activity[8] (Fig. 4a). Therefore, we performed a malachite-green based ATPase assay and measured phosphate release in the presence of ATP (Fig. 4b) using electrophoretically pure PspA (Extended Data Fig. 4a). The ATPase activity of PspA was approximately ten times lower than

bona fide AAA+ ATPases, such as *E. coli* PspF (Extended Data Fig. 4b). The ATPase activity was not significantly reduced in the absence of $Mg^{2+}$ or even in the presence of EDTA, but could be decreased by high concentrations of ATPase inhibitors such as ATPγS or AMPPCP[28]. Measuring the activity upon mutating residues R44, E126 and E179 revealed that these residues are critical for the PspA ATPase activity, as found for Vipp1. Single mutations decreased the ATPase activity by 60–70% and double residue mutations led to a decrease of ~80%. Simultaneous mutation of all three residues reduced the activity by ~90%. By contrast, PspA α1–α5; that is, PspA with truncated helix α0 and the first

**Table 1 | Data collection, image processing and model refinement/validation on sample PspA + ATP**

| **Data collection** | | | | | | | | | | | |
|---|---|---|---|---|---|---|---|---|---|---|---|
| Diameter of structure | 180 Å | 200 Å | 215 Å | 235 Å | 250 Å | 270 Å | 280 Å | 290 Å | 305 Å | 320 Å | 365 Å |
| Movies | 4,470 | 4,470 | 4,470 | 4,470 | 4,470 | 4,470 | 4,470 | 4,470 | 4,470 | 4,470 | 4,470 |
| Magnification | 63,000 | 63,000 | 63,000 | 63,000 | 63,000 | 63,000 | 63,000 | 63,000 | 63,000 | 63,000 | 63,000 |
| Voltage (kV) | 200 | 200 | 200 | 200 | 200 | 200 | 200 | 200 | 200 | 200 | 200 |
| Total dose (e⁻/ Å²) | 58 | 58 | 58 | 58 | 58 | 58 | 58 | 58 | 58 | 58 | 58 |
| Defocus range (µm) | 1.0–4.0 | 1.0–4.0 | 1.0–4.0 | 1.0–4.0 | 1.0–4.0 | 1.0–4.0 | 1.0–4.0 | 1.0–4.0 | 1.0–4.0 | 1.0–4.0 | 1.0–4.0 |
| Physical pixel size (Å) | 1.362 | 1.362 | 1.362 | 1.362 | 1.362 | 1.362 | 1.362 | 1.362 | 1.362 | 1.362 | 1.362 |
| Detector | Gatan K3 | Gatan K3 | Gatan K3 | Gatan K3 | Gatan K3 | Gatan K3 | Gatan K3 | Gatan K3 | Gatan K3 | Gatan K3 | Gatan K3 |
| **Image processing** | | | | | | | | | | | |
| Symmetry imposed | C1, 3.18 Å, −161.1° | C2, 5.75 Å, 34.4° | C1, 2.52 Å, 130.2° | C3, 6.97 Å, 28.3° | C1, 2.10 Å, −83.6° | C2, 3.92 Å, −78.0° | C3, 5.64 Å, 22.9° | C4, 6.92 Å, 21.2° | C1, 1.64 Å, −85.0° | C1, 1.53 Å, −140.3° | C3, 3.95 Å, −52.0° |
| Final no. of segments (ASUs) | 34,695 (659,205) | 64,979 (649,790) | 48,854 (1,172,496) | 46,597 (419,373) | 55,285 (1,603,265) | 63,854 (957,810) | 67,654 (744,194) | 44,377 (399,393) | 25,901 (958,337) | 18,973 (739,947) | 17,663 (264,945) |
| Global map resolution (Å, FSC = 0.143) | 4.1 | 3.8 | 3.9 | 4.4 | 4.5 | 4.4 | 5.4 | 4.7 | 6.6 | 6.9 | 6.9 |
| Local map resolution range (Å, FSC = 0.5) | 4.3–7.6 | 3.8–4.7 | 4.3–6.0 | 5.0–6.3 | 5.0–5.7 | 5.1–6.3 | 5.7–7.0 | 5.5–6.7 | 6.8–8.5 | 7.3–9.1 | 7.4–12.6 |
| Initial model used (PDB code) | 7ABK | 7ABK | 7ABK | 7ABK | 7ABK | 7ABK | 7ABK | 7ABK | 7ABK | 7ABK | 7ABK |
| **Model refinement** | | | | | | | | | | | |
| Model resolution | 4.1 | 3.8 | 3.9 | 4.4 | 4.4 | 4.4 | 5.3 | 4.6 | 6.6 | 6.8 | 6.9 |
| CC mask | 0.79 | 0.82 | 0.79 | 0.83 | 0.86 | 0.79 | 0.73 | 0.79 | 0.80 | 0.79 | 0.81 |
| CC box | 0.86 | 0.87 | 0.85 | 0.89 | 0.91 | 0.87 | 0.81 | 0.87 | 0.85 | 0.85 | 0.87 |
| CC peaks | 0.77 | 0.78 | 0.76 | 0.76 | 0.78 | 0.73 | 0.65 | 0.72 | 0.67 | 0.64 | 0.63 |
| CC volume | 0.80 | 0.82 | 0.80 | 0.83 | 0.85 | 0.78 | 0.74 | 0.79 | 0.80 | 0.79 | 0.80 |
| CC ligands | 0.64 | 0.64 | 0.60 | 0.65 | – | – | – | – | – | – | – |
| Map sharpening B-factor (Å²) | 106.50 | 387.39 | 138.13 | 225.92 | 178.75 | 254.80 | 290.77 | 237.30 | 256.31 | 312.34 | 258.70 |
| **Model composition** | | | | | | | | | | | |
| Nonhydrogen atoms | 96,300 | 96,300 | 96,300 | 96,300 | 94,680 | 94,680 | 94,680 | 94,680 | 94,680 | 94,680 | 94,680 |
| Protein residues | 11,760 | 11,760 | 11,760 | 11,760 | 11,760 | 11,760 | 11,760 | 11,760 | 11,760 | 11,760 | 11,760 |
| **RMSDs** | | | | | | | | | | | |
| Bond lengths (Å) | 0.002 | 0.003 | 0.002 | 0.004 | 0.003 | 0.002 | 0.002 | 0.002 | 0.002 | 0.002 | 0.002 |
| Bond angles (°) | 0.545 | 0.748 | 0.585 | 0.614 | 0.466 | 0.533 | 0.414 | 0.438 | 0.398 | 0.465 | 0.508 |
| **Validation** | | | | | | | | | | | |
| MolProbity score | 1.32 | 1.32 | 1.16 | 1.58 | 1.93 | 1.42 | 1.57 | 1.43 | 1.63 | 1.68 | 1.55 |
| Clashscore | 5.78 | 5.92 | 3.70 | 11.71 | 15.37 | 7.60 | 8.37 | 7.91 | 13.10 | 14.97 | 10.82 |
| Rotamer outliers (%) | 0.00 | 0.13 | 0.00 | 0.00 | 0.00 | 0.00 | 0.61 | 0.61 | 0.61 | 0.00 | 0.00 |
| **Ramachandran plot** | | | | | | | | | | | |
| Favored (%) | 100.00 | 99.15 | 99.48 | 98.45 | 96.39 | 98.45 | 97.42 | 98.45 | 98.97 | 98.45 | 98.97 |
| Allowed (%) | 0.00 | 0.85 | 0.52 | 1.55 | 3.61 | 1.55 | 2.58 | 1.55 | 1.03 | 1.55 | 1.03 |
| Disallowed (%) | 0.00 | 0.00 | 0.00 | 0.00 | 0.00 | 0.00 | 0.00 | 0.00 | 0.00 | 0.00 | 0.00 |
| **Deposition IDs** | | | | | | | | | | | |
| EMDB | 15489 | 15490 | 15491 | 15492 | 15493 | 15494 | 15495 | 15497 | 15496 | 15498 | 15499 |
| PDB | 8AKQ | 8AKR | 8AKS | 8AKT | 8AKU | 8AKV | 8AKW | 8AKY | 8AKX | 8AKZ | 8AL0 |

22 amino acids removed that are far away from the putative active site, did not significantly reduce the ATPase activity. To verify ADP release after ATP hydrolysis, we performed an ADP-Glo assay and measured an ATPase activity of 3 h⁻¹, in agreement with the phosphate release assay (Fig. 4c). After incubation with lipid membranes, the ATPase activity was increased by ~210%. To further investigate the mode of

ATP hydrolysis by PspA, we used mass spectrometry to verify that PspA rods can bind and hydrolyze ATP. Indeed, after incubation of PspA with ATP and extraction of the protein-bound nucleotides, we identified two peaks by high-performance liquid chromatography–tandem mass spectrometry (HPLC–MS/MS) with *m/z* ratios corresponding to ADP and ATP (Fig. 4d). Interestingly, traces of ADP and ATP have also

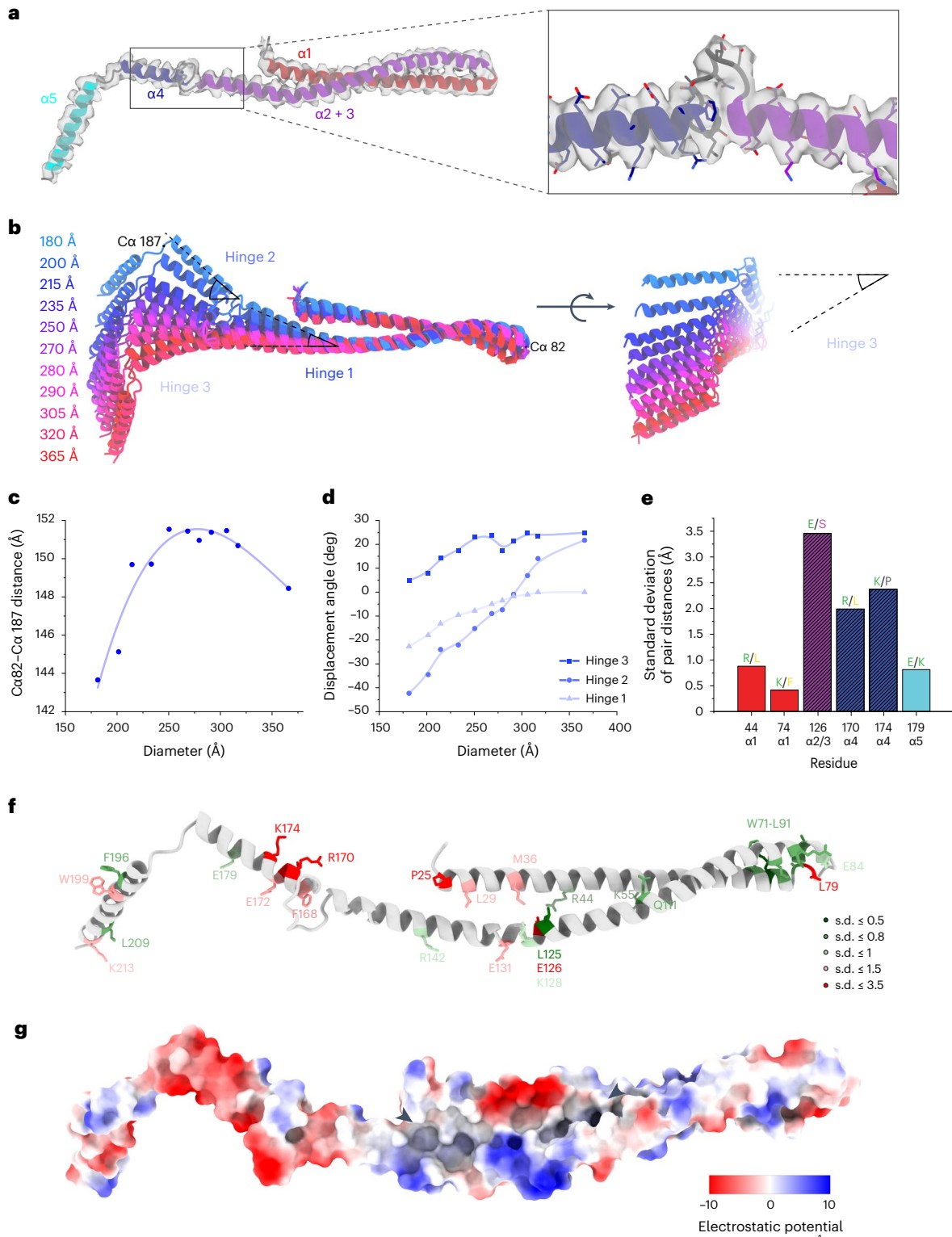

**Fig. 3 | Three hinge regions enable the conformational plasticity of PspA.**
**a**, Model of a PspA + ATP monomer showing details of the density at the loop connecting α3 and α4 (α1, red; α2 + 3, violet; α4, blue; α5, cyan) **b**, Superimposed PspA + ATP monomer structures aligned on the hairpin between α1 and α2 + 3 and color-coded by increasing rod diameter (blue to red for increasing diameters). Hinges 1–3 and the corresponding displacement angle are indicated by opening angle symbols (black). **c**, Graph of PspA + ATP rod diameters over the distance between Cα 82 and Cα 187. **d**, Graph of PspA + ATP rod diameter against the displacement angle at Hinges 1–3. **e**, Bar plot of selected pair distances evaluated over all determined diameter assemblies. For a plot of all evolutionarily conserved residue pairs and a more detailed explanation, see Extended Data Fig. 3c. Color code for residue letters: green, charged; yellow, hydrophobic; pink, polar + uncharged; gray, special cases. Color code for boxes: α1, red; α2 + 3, violet; α4, blue; α5, cyan. **f**, Model of PspA + ATP monomer with evolutionarily conserved amino acid residues highlighted and colored according to the flexibility of their contacts (green, low flexibility (s.d. < 1); red, high flexibility (s.d. > 1)). **g**, Model of the PspA + ATP monomer with electrostatic surface coloring. Arrowheads show the start and end of the non-conserved hydrophobic groove.

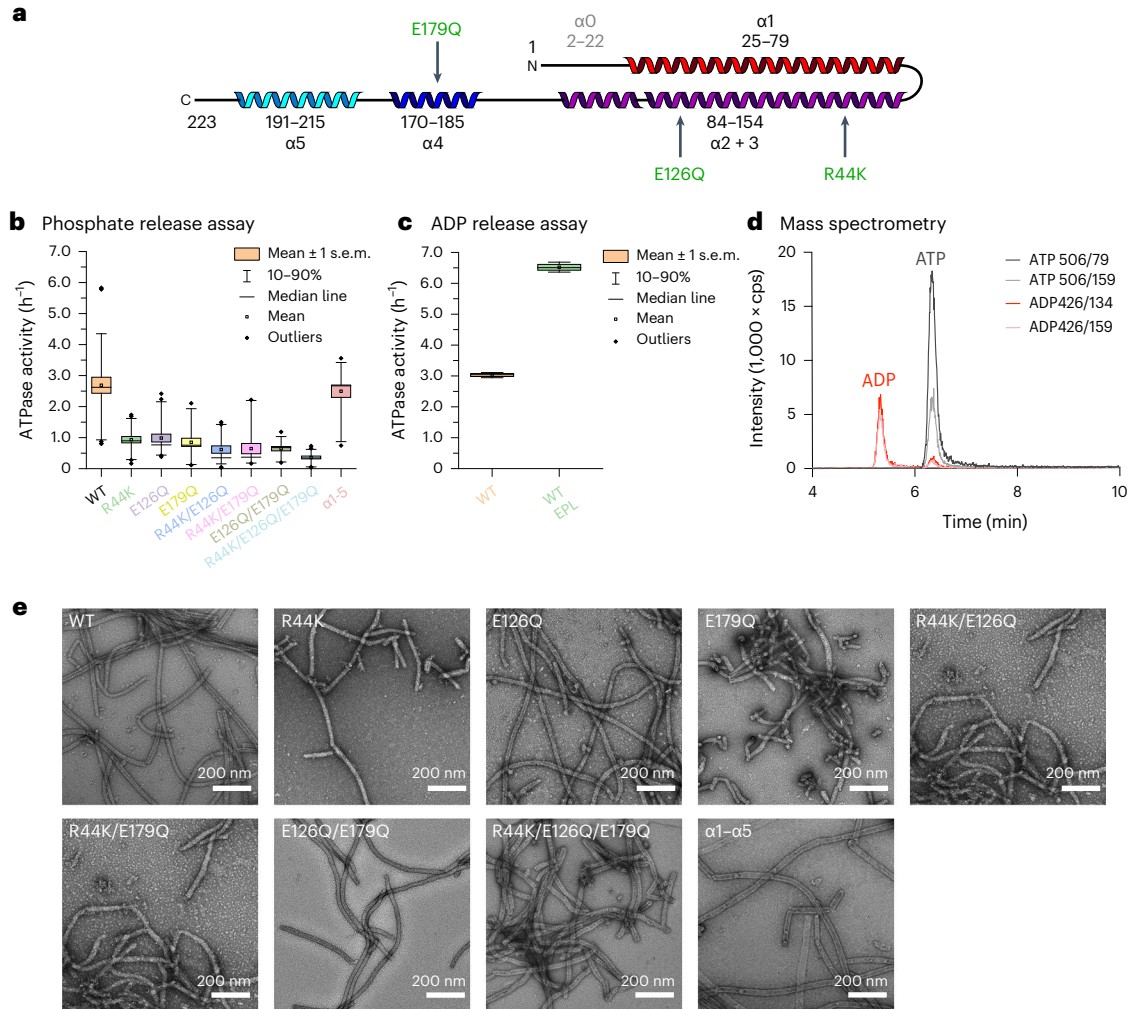

**Fig. 4 | Nucleotide binding and hydrolysis by PspA. a,** Secondary structure topology of the PspA monomer (α1, red; α2 + 3, violet; α4, blue; α5, cyan). Relevant mutations are indicated in green. **b,** ATPase activity of PspA wild type (WT) and mutants measured by a malachite-green-based assay. The boxplots show the mean ± s.e.m. as boxes, the 10–90th percentile as whiskers and outliers as diamonds. *P* values of two-sample *t*-test: WT versus R44K/E126Q/E179Q, $P = 7.18 \times 10^{-9}$; WT versus α1–α5, $P = 0.63$. WT, $n = 36$; R44K, $n = 24$; E126Q, $n = 27$; E179Q, $n = 18$; R44K/E126Q, $n = 21$; R44K/E179Q, $n = 18$; E126Q/E179Q, $n = 18$; R44K/E126Q/E179Q, $n = 21$; α1–α5, $n = 18$. For all measurements, samples of at least three biological replicates were included. **c,** ATPase activity of the WT protein and the α1–α5 mutant in the absence and presence of EPL (SUVs made

from *E. coli* polar lipid extract). The boxplots show the mean ± s.e.m. as boxes, the 10–90th percentile as whiskers and outliers as diamonds. WT (orange), ATPase activity in the absence of EPL; WT EPL (green), ATPase activity in the presence of EPL; α1–α5 variant (purple); $n = 3$. For all measurements, samples of at least three biological replicates were included. **d,** HPLC–MS/MS of PspA + ATP after 24 h incubation and extensive washing. Different color lines represent different multiple reaction-monitoring (MRM) transitions. For ADP, MRM transitions are 426/134 (red) and 426/159 (light red). For ATP, MRM transitions are 506/79 (dark gray) and 506/159 (light gray). The ADP fragments below the ATP peak are formed by in-source fragmentation of ATP in the ESI source. **e,** Negative staining EM micrographs of PspA WT protein and PspA mutants. Magnification, ×57,000.

been found in a PspA sample without the addition of ATP, indicating that ADP and ATP were routinely co-purified from the expression host (Extended Data Fig. 4c). In agreement with our phosphate release measurements, HPLC–MS/MS indicated that the R44K/E126Q/E179Q mutant was still capable of ATP binding and hydrolysis but only to about half of the wild-type protein (Extended Data Fig. 4d,e). The tested mutants were still capable of forming large rod structures comparable with the wild-type protein (Fig. 4e). Biochemical and mass spectrometry analysis of PspA confirmed the conservation of ATPase activity in bacterial PspA and Vipp1 proteins. We showed that PspA can bind and hydrolyze ATP, albeit with an order of magnitude lower activity than observed for canonical P-loop or Walker A motif ATPases.

## ATP enhances PspA-induced membrane remodeling

To assess the consequences of the structural plasticity of PspA in the context of membrane remodeling, we analyzed the cryo-images for

structural and morphological changes of added *E. coli* polar lipid (EPL) small unilamellar vesicles (SUVs) and the effect of ATP under reconstitution conditions developed for optimal tubulation (50 nm SUVs and in situ refolding) (Fig. 5a). After incubation with PspA, we observed PspA rods engulfing and tubulating membranes (Fig. 5b). Notably, PspA rods engulfing membranes did not have uniform diameters. Instead, they were frequently attached to vesicles with one wider end tapering towards the distal end. Moreover, engulfed membranes were continuous lipid tubes as well as isolated small vesicles and membrane patches. In the absence of PspA, control SUVs had a narrow bilayer thickness distribution from 27 Å to 32 Å with a mean bilayer thickness of 31 Å (Fig. 5c). After incubation with PspA, SUV membranes were significantly thicker on average, with a broad distribution and a major peak at 38 Å and a minor peak at 68 Å. In the presence of PspA and ATP, the bilayer thickness distribution closely resembles the PspA sample (mean thickness, 37 Å). The observed increase in bilayer thickness

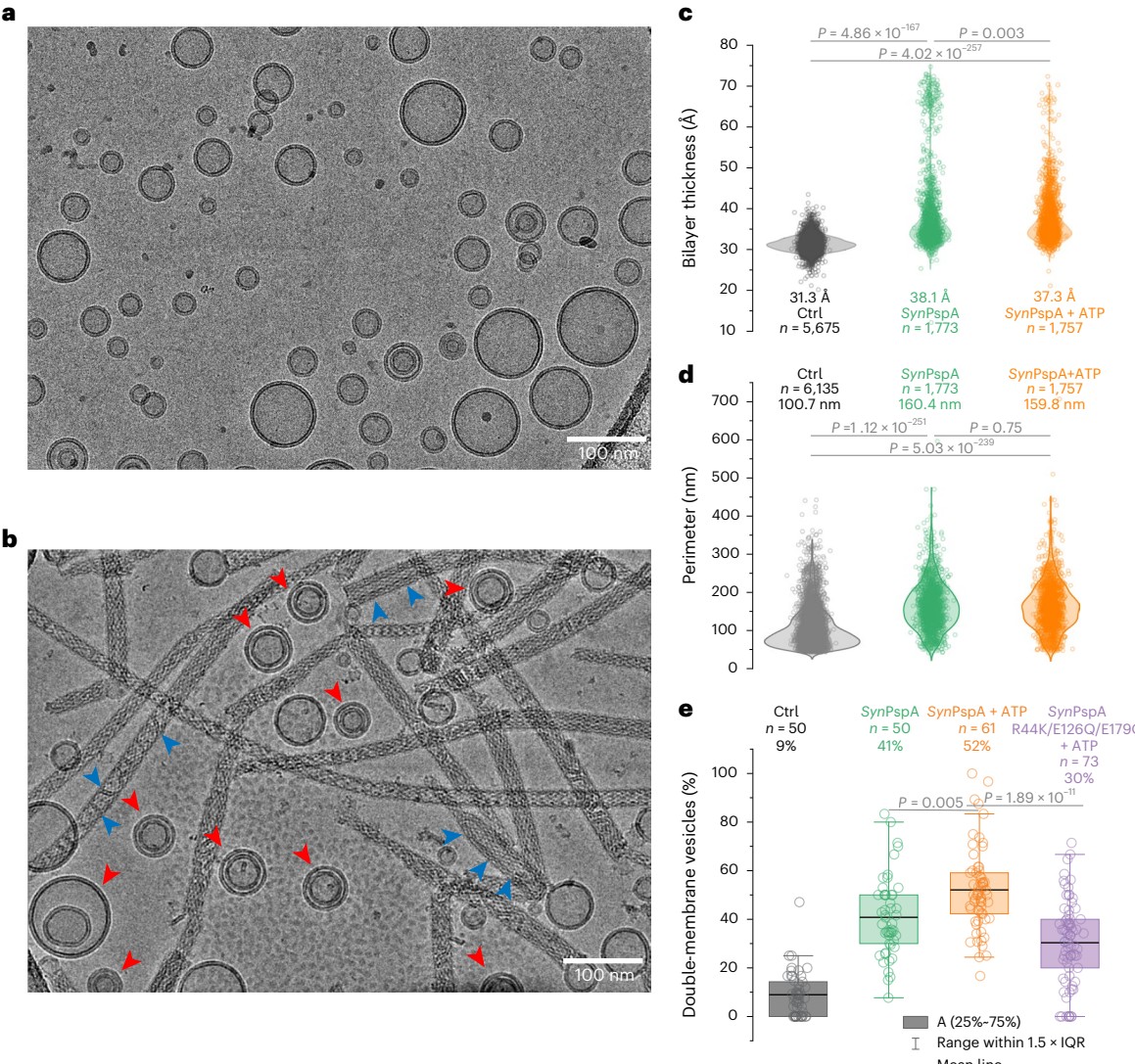

**Fig. 5 | Changes of EPL vesicle morphologies after _Syn_PspA ± ATP reconstitution. a**, Selected cryo-EM micrograph from 632 recorded movies of EPL control vesicles. **b**, Selected cryo-EM micrograph from 50 recorded movies of the PspA + EPL sample. Red arrowheads indicate encapsulated vesicles. Blue arrowheads indicate engulfed membranes. **c**, Violin plot of the bilayer thickness of control EPL SUVs (gray), PspA + EPL SUVs (green) and PspA + EPL + ATP SUVs (orange). The mean bilayer thickness, the number of measured vesicles (_n_) and _P_ values from a two-sample _t_-test are indicated on the graph. **d**, Violin plot of the vesicle perimeter of control EPL SUVs (gray), PspA + EPL SUVs (green) and PspA + EPL + ATP SUVs (orange). The mean vesicle perimeter, the number of

measured vesicles (_n_) and _P_ values from a two-sample _t_-test are indicated on the graph. **e**, Box plot of the relative occurrence of double-membrane vesicles in each sample: control EPL SUVs (gray), PspA + EPL SUVs (green), PspA + EPL + ATP SUVs (orange) and PspA (R44K, E126Q, E179Q) + EPL + ATP SUVs (violet). The relative number of double-membrane vesicles, the number of measured vesicles (_n_) and _P_ values from a two-sample _t_-test are indicated on the graph (Ctrl versus _Syn_PspA, $P = 1.83 \times 10^{-19}$; Ctrl versus _Syn_PspA + ATP, $P = 1.13 \times 10^{-31}$; Ctrl versus _Syn_PspA R44K/E126Q/E179Q + ATP, $P = 1.24 \times 10^{-14}$; _Syn_PspA versus _Syn_PspA + ATP, $P = 0.005$; _Syn_PspA + ATP versus _Syn_PspA R44K/E126Q/E179Q + ATP, $P = 1.89 \times 10^{-11}$). IQR, interquartile range.

after incubation with PspA is consistent with a previous report[9]. To monitor changes in vesicle sizes, we used the vesicle perimeter as a descriptor (Fig. 5d and Extended Data Fig. 5a). The EPL SUVs alone had a mean perimeter of 100 nm, corresponding to a mean diameter of 32 nm. Similar to the bilayer thickness, the distribution of vesicle perimeters was significantly increased in the presence of PspA and PspA + ATP, peaking at 160 nm and 150 nm, respectively. The vesicle size increase observed after incubation with PspA agrees with the previously reported PspA-mediated vesicle fusion[9]. Another prominent feature of the vesicles containing PspA was the formation of small, double-membrane vesicles (that is, small vesicles that are encapsulated by a larger vesicle), in analogous topology to intra-luminal vesicles (ILVs) (Fig. 5b, red arrowheads). Approximately 9% of all vesicles were double-membrane vesicles in the control, whereas the share of

double-membrane vesicles increased in the PspA preparations to 41% and was highest in the PspA + ATP sample at 52% (Fig. 5e). Using the ATPase-deficient mutant PspA R44K/E126Q/E179Q as a negative control, we validated that indeed it was the ATPase activity of PspA and not the addition of ATP that gave rise to the increased number of double vesicles in the PspA + ATP sample. In addition, the distance between the two enclosed vesicle membranes, termed here the enclosure distance, was on average higher in the control vesicles (>70 Å) with a very broad enclosure distance distribution (Extended Data Fig. 5b,c). Both the PspA as well as the PspA + ATP sample showed a well-defined distance distribution with a mean enclosure distance of 67 Å and 62 Å, respectively. Compared with the vesicle perimeter, only the smallest vesicles (perimeter of 100–200 nm) showed this constant enclosure distance of ~50–70 Å (Extended Data Fig. 5d). Together, this quantitative image

analysis of vesicle characteristics showed that the presence of ATP increases the occurrence of double-membrane vesicles through the enhanced membrane remodeling capabilities of PspA.

### Effect of ATP analogs and mutations on PspA's plasticity

To further investigate how nucleotide binding and hydrolysis affects the PspA rods, we analyzed the rod diameter distribution after the addition of ATP to preformed PspA rods using cryo-EM. The resulting rod distribution showed two distinct peaks at 215 Å and 250 Å diameters, whereas only a minor shift to larger diameters compared with the rod distribution after the addition of ATP during rod assembly was observed (Extended Data Fig. 6a,b). These observations indicate that only minor diameter changes are possible once the rods have formed, and larger changes can only occur during rod assembly or re-assembly. Additionally, we tested the effect of the non-hydrolyzable ATP analogs AMPPCP and ATPγS on the diameter distribution using cryo-EM. First, upon addition of AMPPCP during rod formation, we found that the diameter distribution equals the distribution in the presence of ADP supported by the average ± s.d. measurements: 240 ± 40 Å and 240 ± 50 Å of the ADP and the AMPPCP distribution, respectively (Extended Data Fig. 6c,d). Unlike canonical ATPases[29], these data indicate that AMPPCP mimics the ADP-bound state in the case of PspA. Second, upon the addition of ATPγS, the diameter distribution overlapped highly with the ATP sample, although that showed an additional tail of higher diameter rods (>300 Å) (Extended Data Fig. 6e,f). Assuming that ATPγS mimics the ATP transition state, this observation suggests that complete ATP turnover rather than mere binding is required for the formation of rods with large diameters.

In the series of PspA structures, we found the second hinge, that is, the loop connecting α3 and α4 (loop α3/α4), to be critically important for the conformational adjustments of PspA monomers in rods with higher diameters and also participating in the putative nucleotide binding region. Therefore, we generated a PspA mutant lacking ten residues of the loop E156–S165 (PspA dL10) (Fig. 6a) and analyzed the ATPase activity as well as diameter distribution. First, PspA dL10 showed an ATPase activity that was reduced by over 90% compared with wild-type PspA, although the residues critical for the ATPase activity (R44, E126, E179) were still intact (Fig. 6b). Second, we determined the structure of PspA dL10 and compared it with the wild-type protein. We found that PspA dL10 still formed rods with symmetries and diameters also present in the wild-type sample, although with a narrower distribution centered at slightly higher diameters (Fig. 6c,d). Upon addition of ATP during the formation of PspA dL10 rods, we did not observe significant differences in the diameter distribution of the determined structures (220 ± 20 Å (dL10) versus 230 ± 30 Å (dL10 ATP)), in agreement with the highly reduced ATPase activity (Fig. 6e and Extended Data Fig. 6g,h). The structures of PspA dL10 rods were almost identical to the wild-type rods, but the density for the loop between α3 and α4 was missing (Fig. 6f). Together, the dL10 rod structures indicate that loop α3/α4 is not essential for the formation of PspA rods per se but is critical for the formation of rods with very low and high diameters as well as for the associated ATPase activity. By removing the loop α3/α4, we reduced the conformational freedom, providing an efficient means to restrict the observed structural plasticity of PspA assemblies.

## Discussion

We revealed in molecular detail the structural plasticity of PspA that gives rise to a large series of related assembly structures. We determined a total of 61 cryo-EM structures of helically assembled PspA rods spanning rod diameters from 180 Å to 400 Å under different conditions: in the absence of nucleotides, and in the presence of ADP and ATP, respectively (Figs. 1 and 2). We found that the distribution of PspA rod diameters is affected by nucleotides, resulting in the largest rod diameters and widest diameter range after incubation with ATP, intermediate-sized diameter rods with narrow distribution with ADP,

and, in the absence of nucleotides, small diameter rods with intermediate distribution. We obtained 14 unique assembly structures over the five samples including non-hydrolyzable ATP analogs AMPPCP and ATPγS. The diameters changes were caused by conformational changes primarily at three hinge regions between the α-helical segments of PspA (Fig. 3). Moreover, we biochemically characterized the ATPase activity of PspA and studied mutants abrogating the hydrolytic activity while maintaining rod formation (Fig. 4). To link the structural plasticity and ATPase activity of PspA with a biochemical function, we analyzed changes in the membrane vesicle morphology induced by PspA in the presence and absence of ATP and found that ATP appears to enhance the membrane remodeling capabilities of PspA (Fig. 5). Structural and biochemical analysis of a mutant lacking the loop connecting α3 and α4 revealed that the second hinge region is essential for the structural plasticity and ATPase activity of PspA (Fig. 6).

Superimposing the now-determined structures, we identified three hinge regions that enable PspA to flex from a high-tension bow shape over a low-tension linear shape back to a high-tension inverted bow shape. These changes expose slightly modified subunit interfaces and, as a result, PspA's mass packing per unit length of polymer increases. Ultimately, these packing changes result in increasing polymer diameters, from the observed lowest diameters of 180 Å to the highest observed diameters up to 400 Å. As higher diameter rods only formed in the presence of ADP or ATP, the monomer tension in these rods may be stabilized in the process of ATP binding or hydrolysis, presumably as a result of the formation of new interactions between protein and the nucleotide. Interestingly, the putative ATP or ADP binding pocket is formed by four PspA chains ($j_0$, $j_{-1}$, $j_{+26}$, and $j_{+27}$) and α1–α4 as well as the respective loop regions. Relevant catalytic residues, including residues of the α3–α4 connecting loop point toward the potential binding site and were found to be solvent-accessible. The three hinge regions are close to the putative nucleotide binding residues in the assembly. According to the weak density in some but not all cryo-EM reconstructions and in reference to the Vipp1 structure[8], it is likely that not every binding site is equally occupied by nucleotides. In the Vipp1 ring structures, ADP was found in one discrete Vipp1 layer: the most constricted layer of the ring. The presence of ADP in this single layer suggests that only a fraction of putative sites correspond to be active in one ring assembly. The low ATPase activity of *Synechocystis* PspA is in accordance with the reported GTPase and ATPase activity of the closely related *E. coli* Vipp1 and PspA[8,25–27]. In support, low competition rates with non-hydrolyzing ATP analogs reveal differences in ATP turnover when compared with canonical P-loop or Walker A motif ATPases. However, when few layers per rod molecule are considered actively hydrolyzing, the effective turnover can be much higher locally. For example, considering that the average length of PspA rods is 2.6 ± 1.3 μm, a total of approximately 10,000 monomers are found in the rod, corresponding to 1,000 turns depending on the symmetry. Therefore, we reason that local ATP turnover can be approximately 1,000 times higher; that is, three orders of magnitude, resulting in apparent 50 ATP molecules per minute (3,000 h⁻¹), which is in the determined activity range of actively remodeling dynamin[30].

Interestingly, ATP hydrolysis is critical for the disassembly or assembly of eukaryotic ESCRT-III polymers mediated by the dedicated AAA+ ATPase Vps4 (refs. 31,32). By hydrolyzing ATP, Vps4 can remove and unfold individual ESCRT-III subunits from the polymer structures[33]. Although some Psp systems also include an AAA+ ATPase (PspF), it does not appear to be essential for the membrane remodeling properties of PspA, as not all organisms that contain PspA also contain PspF (for example, *Synechocystis* sp. PCC 6803 does not). Vps4 is thought to recruit ESCRT-III members through its amino-terminal microtubule interacting and trafficking domain[34,35]. Vps4 triggers a conformational change in ESCRT-III polymers by exchanging subunits of the filament polymers to allow greater curvature, which can cause the tubes to constrict[36]. This conformational change can lead to the fission of

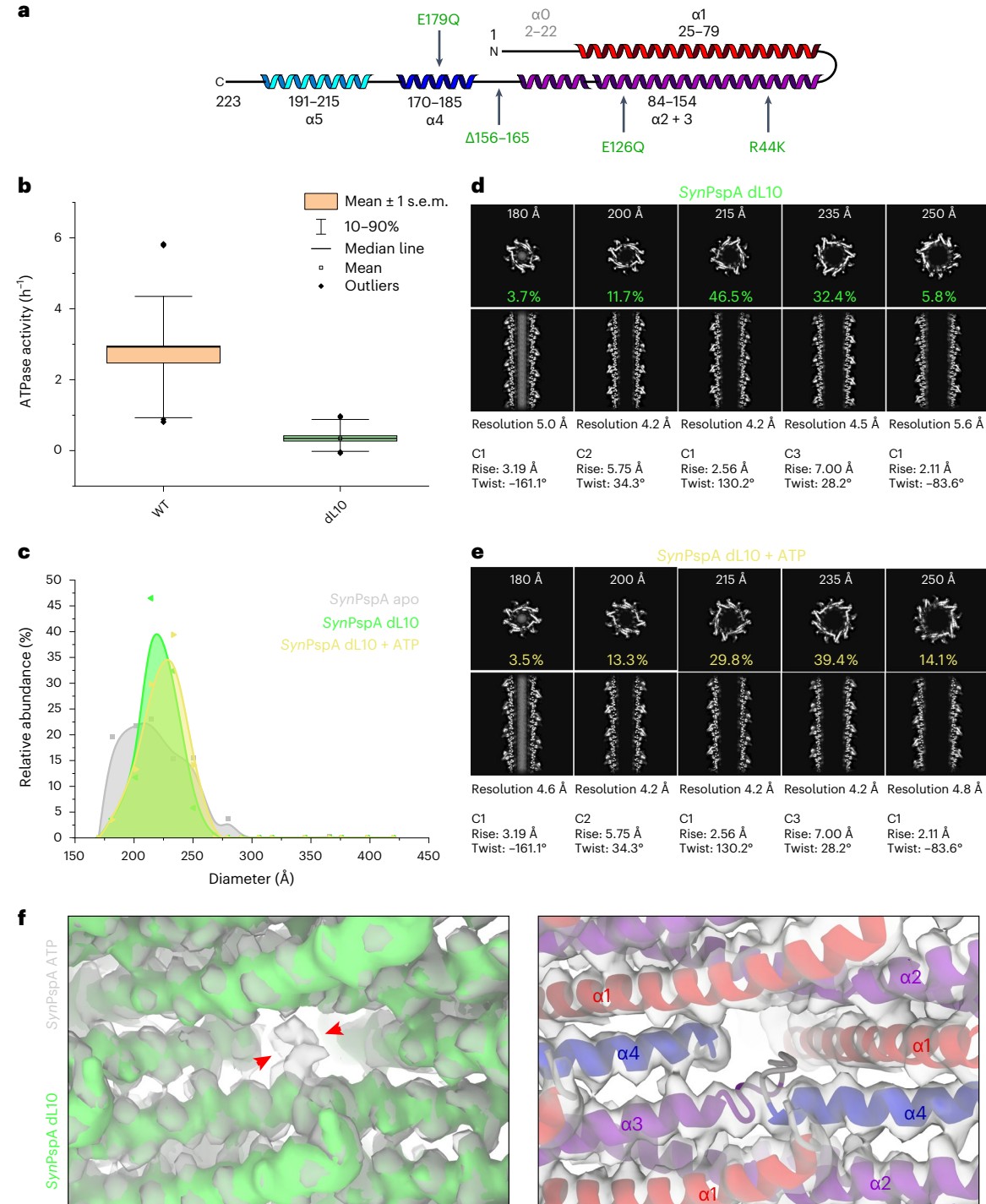

**Fig. 6 | ATPase activity and structure of the PspA dL10 mutant. a**, Schematic secondary structure topology of the PspA monomer (α1, red; α2 + 3, violet; α4, blue; α5, cyan). The deletion of loop residues 156–165 (dL10) and relevant mutations are indicated in green. **b**, ATPase activity of PspA WT and dL10 measured by a malachite-green-based assay. The boxplots show the mean ± s.e.m. as boxes, the 10–90th percentile as whiskers and outliers as diamonds. $P$ values of two-sample $t$-test: WT versus dL10, $P = 3.92 \times 10^{-10}$; WT, $n = 39$, dL10, $n = 23$. For all measurements, samples of at least three biological replicates were included **c**, Histogram of rod diameters and the normalized

abundance of PspA dL10 rods (green), PspA dL10 rods in the presence of ATP (yellow) and PspA WT rods (gray) for comparison. PspA dL10 has a narrow diameter distribution that is not affected by ATP addition. **d,e**, The associated cryo-EM structures of PspA dL10 (**d**) and PspA dL10 + ATP (**e**) are shown in cross-sectional top and side views. **f**, Left, comparison of the cryo-EM density maps of PspA + ATP (gray) and PspA dL10 (green). Red arrowheads indicate the additional density for the loop that is missing in the dL10 map. Right, cryo-EM density map of PspA dL10 with the PspA WT model.

bound vesicles by decreasing filament diameter in membrane-bound complexes[37]. For initial ILV biogenesis, the complete ESCRT-III heteropolymer structure is assembled, and after binding of Vps4, membrane

constriction and release of the ILVs can occur[38]. It remains to be established whether the observed modulation of PspA rod diameters through ATP and ADP binding and/or hydrolysis is of physiological

relevance in the bacterial cell. PspA rods mixed with liposomes produced double-membrane vesicles that share the same topology as ILVs and are likely a result of inward-vesicle budding[9]. In accordance with our previous work, we suggest that double-membrane vesicles are formed by cross-membrane vesicle transfer; that is, by releasing of PspA-engulfed membrane into an acceptor vesicle. Here, we demonstrated that after PspA incubation, the number of double-membrane vesicles increased in the presence of ATP. This observation indicates that the membrane remodeling activity of PspA is stimulated by ATP, by facilitating the uptake of vesicle membranes when larger diameter rods engulf membranes more efficiently than smaller diameter rods owing to lowered membrane curvature. Therefore, our observations provide a direct link between the elusive NTPase activity of the Vipp1/PspA protein family[8,25–27,39] with an enhanced production of ILV-like vesicles.

For eukaryotic ESCRT-III proteins, Vps4 has been proposed to mediate constriction and membrane fission accompanied by a reduction in the diameter of the polymeric structures through the hydrolysis of ATP[40,41], in analogy to dynamin, which accomplishes constriction through GTP hydrolysis. By contrast, ATP addition during PspA tube formation leads to dilation of rod diameters. The results of ATP incubation with preformed PspA rods suggest that large diameter changes of PspA rods can apparently only occur during rod assembly or re-assembly, which may be accomplished by constant disassembly and assembly by chaperones or PspF in the bacterial cell, in analogy to Vps4 in eukaryotic cells. Although it cannot be distinguished from our data whether ATP binding and/or hydrolysis leads to the dilation, or conversely, whether ADP and ATP release leads to rod constriction, the structural similarity between PspA and the ESCRT-III proteins raises the question of whether other ESCRT-III proteins also possess a nucleotide binding pocket like that of PspA and therefore have intrinsic ATPase activity. Reports are emerging that eukaryotic as well as archaeal ESCRT-III proteins can bind DNA or have been localized in foci close to chromatin[42–44]. A cryo-EM structure of an IST1/CHIMP1B polymer bound to negatively charged DNA revealed a binding site at the inner lumen of the tubes in potential competition with the negatively charged lipid membrane[43]. By contrast, the here-identified putative ATP binding pocket in PspA polymers is located centrally in the tube wall, sandwiched between adjacent subunits. Together, a common feature of bacterial and eukaryotic ESCRT-III proteins (together with associated ATPases such as Vps4) is the ability to bind nucleotides and exhibit different conformational states of the monomers.

In this study, we describe how a combination of fixed and switching residue interactions mediate the observed structural plasticity of PspA. The fixed residues are of polar as well as hydrophobic nature and probably serve as anchor points to keep the polymer assembled. Switching residues include evolutionarily non-conserved hydrophobic residues forming a groove on the α1–α2 hairpin, which are responsible for establishing the required structural flexibility of assembling different diameter rods. Defined residue contacts also explain why discrete diameter states were observed as opposed to an ensemble of quasi-continuous changes of diameters. The observed structural plasticity is a common property of several polymeric protein assemblies in various biological fields. For instance, structural rearrangements were identified in early fundamental work on tobacco mosaic virus coat protein, in which pH and ionic strength were found to determine the type of disc, stacked discs and helix assemblies[45]. Structural plasticity has also been observed in a series of lipid remodeling complexes; that is, ANTH/ENTH clathrin adaptor complexes have been shown to form helical assemblies over a range of different diameters, each varying by an additional subunit in the helical turn[46]. Notably, eukaryotic ESCRT-III proteins have been shown to portray structural polymorphism, assembling in sheets, strings, rings, filaments, tubules, domes, coils and spirals[4–7]. The structural changes observed now for the hinges connecting α3 and α4 as well as α2 and α3 are consistent with structures of 180 Å CHMP1B/IST1 copolymer and 280 Å diameter CHMP1B filaments showing IST1 binding-induced conformational changes in the homologous hinge regions of the ESCRT-III protein CHMP1B[24]. Similarly, tubular assemblies with varying diameters were observed with human ESCRT-III CHMP2A/CHMP3 assemblies, and it has been hypothesized that such plasticity is involved in filament sliding that could drive membrane constriction[23]. A related membrane remodeling molecule, human dynamin 1, has been shown to exhibit large-scale GTPase-mediated structural rearrangements accompanied by a change in assembly diameter[47]. In the case of the canonical dynamin, GTP binds to the dedicated catalytic G-domain and enables filament constriction for the final membrane fission step. By contrast, PspA binds and hydrolyzes ATPs between the subunits of the polymer, presumably causing a dilation of the rods. However, for dynamin family member EHD1, it has been shown that ATP hydrolysis mediates membrane remodeling by dilation of membrane tubulation EHD1 scaffolds[48]. Although the structural plasticity observed here may be crucial for PspA membrane remodeling activity[9], it remains to be seen whether the small-scale conformational changes are sufficient to trigger membrane dilation or constriction. Clearly, further characterization is required; in particular, to better understand the structural transitions of PspA assemblies in the presence of lipid membranes.

## Online content

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

## Methods

### Expression and purification of PspA and EcoPspF

PspA wild type (*slr1188*) of *Synechocystis* sp. PCC 6803 and associated mutants (R44K, E126Q, E179Q, R44K/E126Q, R44K/E179Q, E126Q/E179Q, R44K/E126Q/E179Q, α1–α5 (deletion of α0 (amino acids 1–23), dL10 (deletion of amino acids 156–165, the loop between α3 and α4)) were heterologously expressed in *E. coli* C41 cells in TB medium using a pET50(b)-derived plasmid.

For purification of PspA and associated mutants under denaturing conditions, cells were resuspended in lysis buffer containing 6 M urea (10 mM Tris-HCl pH 8.0, 300 mM NaCl) supplemented with a protease inhibitor. Cells were lysed in a cell disruptor (TS Constant Cell disruption systems 1.1 KW; Constant Systems). The crude lysate was supplemented with 0.1% (v/v) Triton X-100 and incubated for 30 min at room temperature (20–25 °C). Subsequently, the lysate was cleared by centrifugation for 15 min at 40,000*g*. The supernatant was applied to Ni-NTA agarose beads. The Ni-NTA matrix was washed with lysis buffer and lysis buffer with additional 10–20 mM imidazole. The protein was eluted from the Ni-NTA with elution buffer (10 mM Tris-HCl pH 8.0, 1,000 mM imidazole, 6 M urea). The fractions containing protein were pooled, concentrated (Amicon Ultra-15 centrifugal filter 10 kDa MWCO) and dialyzed overnight against 10 mM Tris-HCl pH 8.0 or buffer containing 2 mM ADP or ATP with 4 mM MgCl₂ (8 °C, 10 kDa molecular weight cutoff (MWCO)), including three buffer exchanges. Protein concentrations were determined by measuring the absorbance at 280 nm of PspA diluted in 4 M guanidine hydrochloride using the respective molar extinction coefficient calculated by ProtParam[50].

PspF(1–275) of *E. coli* K12 was heterologously expressed in *E. coli* BL21 (DE3) cells in TB medium using a pET50(b)-derived plasmid. For purification of EcoPspF, cells were resuspended in lysis buffer (10 mM Tris-HCl pH 8.0, 5% (v/v) glycerol, 50 mM NaCl) supplemented with a protease inhibitor. Cells were lysed in a cell disruptor (TS Constant Cell disruption systems 1.1 KW; Constant Systems) followed by centrifugation of the lysate for 15 min at 5,000*g*. The supernatant was applied to Ni-NTA agarose beads. The Ni-NTA matrix was washed with lysis buffer and lysis buffer with 20–60 mM imidazole. The protein was eluted from the Ni-NTA with elution buffer (10 mM Tris-HCl pH 8.0, 50 mM NaCl, 1,000 mM imidazole). The fractions containing protein were pooled and dialyzed overnight against dialysis buffer (10 mM Tris-HCl pH 8.0, 5% (v/v) glycerol, 50 mM NaCl, 0.1 mM EDTA, 1 mM dithiothreitol) (8 °C, 10 kDa MWCO), including three buffer exchanges. Protein concentrations were determined by measuring the absorbance at 280 nm of EcoPspA using the respective molar extinction coefficient calculated by ProtParam[50].

### Liposome preparation and lipid handling

Chloroform-dissolved EPL extract was purchased from Avanti Polar Lipids. Lipid films were produced by evaporating the solvent under a gentle stream of nitrogen and vacuum desiccation overnight. The lipid films were rehydrated in 10 mM Tris-HCl pH 8.0 by shaking for 30 min at 37 °C. The resulting liposome solution was subjected to five freeze–thaw cycles combined with sonication at 37 °C in a bath sonicator. SUVs were generated by extrusion of the liposome solution through a porous polycarbonate filter (50 nm pores). For *Syn*PspA lipid reconstitution, unfolded *Syn*PspA (in 6 M urea) or buffer with 6 M urea for the controls was added to EPL SUVs and incubated at room temperature for 15 min. Then the mixture was dialyzed overnight against 10 mM Tris-HCl pH 8.0 (8 °C, 10 kDa MWCO) including three buffer exchanges.

### ATPase activity determination

To test phosphate release from PspA upon ATP hydrolysis, a malachite-green assay was used (PiColorLock Gold Phosphate Detection Kit). PspA (4–8 µM) was mixed with 2 mM ATP and 4 mM MgCl₂ in 10 mM Tris-HCl pH 8.0 and incubated at 37 °C for 60 min. The reaction was stopped by the addition of the malachite-green mixture ('Gold mix') and incubated

for 5 min at room temperature before adding the 'stabilizer'. Samples were transferred to a 96-well plate and incubated for 10 min at 25 °C with shaking. Absorbance at 620 nm with pathlength correction was measured using a Tecan infinite M nano plate reader. To determine the inorganic phosphate (Pᵢ) released by PspA, the absorbance was corrected for ATP and protein background. Then the Pᵢ release was calculated by linear regression from a phosphate standard curve. The ATPase activity of PspA was calculated as:

$$A_{\text{ATPase}} = \frac{\Delta[\text{P}_i]}{[\text{Protein}] \times \Delta t} \qquad (1)$$

where $A_{\text{ATPase}}$ is the average ATPase activity, $\Delta[\text{P}_i]$ is the phosphate release (in µM), [Protein] is the protein concentration (in µM) and $t$ is the incubation time of the reaction (in hours).

To determine the ATPase activity of PspA in the presence of lipids, the ADP formed during the ATPase reaction of PspA was measured using a Promega ADP-Glo Assay. PspA (4–8 µM) reconstituted or incubated with lipids was mixed with 1 mM ATP and 2 mM MgCl₂ in 10 mM Tris-HCl, pH 8.0 and incubated at 37 °C for 60 min. After the reaction, ADP-Glo Reagent was added and the mixture was incubated for 40 min at 25 °C with shaking to deplete the remaining ATP. Finally, Kinase Detection Reagent was added to detect the released ADP by a luciferase reaction. After 60 min reaction time at 25 °C, the mixture was transferred to a 96-well plate, and luminescence was detected using a Bio-Rad MP Gel-Doc. Luminescence intensity was determined densitometrical using ImageJ (v.2.14)[51]. ADP concentrations were calculated by linear regression from an ADP standard curve. The ATPase activity was calculated analogously to Eq. (1).

### Detection and Quantification of ATP and ADP by mass spectrometry

To determine the binding of nucleotides to PspA, mass spectrometric analysis was performed. A 2 ml preparation containing 80 nmol PspA, 2 mM ATP and 4 mM MgCl₂ in 10 mM Tris-HCl buffer, pH 8.0 was incubated at 37 °C (600 rpm) for 24 h. After incubation, the protein was pelleted at 20,000*g* (4 °C) for 15 min. The supernatant was discarded and the protein pellet was washed twice with 2 ml of Tris-HCl buffer, pH 8.0. After the second wash, the nucleotides were extracted using methanol. The sample was precipitated with 1 ml 80% methanol with 10 mM Tris-HCl, pH 8.0 for 30 min at room temperature. The extracted nucleotides were separated from the precipitated protein by centrifugation at 20,000*g* for 15 min (4 °C) followed by storage of the extract at −20 °C. As a control, a similar nucleotide extraction was carried out with protein directly after purification. To account for autohydrolysis of the nucleotides, 2 mM ATP was incubated for 24 h at 37 °C at the same buffer conditions. The autohydrolysis did not exceed 1%.

The supernatants were dried in a SpeedVac (RVC-2-25, Christ) and resuspended in 100 µl 20 mM triethylammonium acetate pH 9.0. For quantification of ATP and ADP, an HPLC–MS/MS method with electrospray ionization (ESI) in negative modus was developed using a Qtrap6500 instrument (ABSciex) coupled with an Agilent 1260 HPLC. Chromatographic separation was performed on a Zorbax Eclipse Plus (4.6 × 100 mm; 2.6 µm particle size; Agilent). The column temperature was kept at 30 °C. The mobile phases consisted of 20 mM triethylammonium acetate, pH 9.0 (solvent A) and 20 mM triethylammonium acetate, pH 9.0 with 20% acetonitrile (solvent B) at a flow rate of 500 µl min⁻¹. The sample injection volume was 10 µl. An isocratic step of 20% solvent B (0–8 min) was followed by an increase to 99% solvent B within 0.1 min, which was held for 5 min for cleaning. The gradient returned to 20% solvent B within 0.1 min and the system was equilibrated for 5 min. Quantitation after HPLC separation was performed using ESI–MS/MS detection in multiple reaction-monitoring mode, which allows monitoring of a specific fragmentation reaction at a given retention time. The parameters used for the quantification of ATP and ADP are listed

in Extended Data Table 1. Mass spectrometer settings were curtain gas ($N_2$), 40 arbitrary units (a.u.); temperature of the source, 500 °C; nebulizer gas ($N_2$), 80 a.u.; heater gas ($N_2$), 40 a.u.; and ion spray voltage, −4500 V. Data acquisition and processing were carried out using the software Analyst v.1.6.1 (ABSciex). For quantification, the software Multiquant (ABSciex) was used. The calibration curve showed linearity between 10 nM and 10 μM ATP and ADP with a correlation coefficient of $R^2 = 0.9993$ (ATP) and $R^2 = 0.9991$ (ADP).

## Negative staining electron microscopy

For negative staining electron microscopy, 3 μl of the sample was applied to glow-discharged (PELCO easiGlow Glow Discharger; Ted Pella) continuous carbon grids (CF-300 Cu; Electron Microscopy Sciences). The sample was incubated on the grid for 1 min. Then the grid was side-blotted using filter paper, washed with 3 μl of water, stained with 3 μl of 2% uranyl acetate for 30 s and air-dried. The grids were imaged with a 120 kV Talos L120C electron microscope (Thermo Fisher Scientific/FEI) equipped with a CETA camera at a pixel size of 2.49 Å per pixel (×54,000 magnification) at a nominal defocus of 1.0–2.5 μm.

## Cryo-EM

PspA grids were prepared by applying 3.5 μl PspA (Extended Data Table 2) to glow-discharged (PELCO easiGlow Glow Discharger; Ted Pella) Quantifoil grids (R1.2/1.3 Cu 200 mesh; Electron Microscopy Sciences). The grids were plunge-frozen in liquid ethane using a Thermo Fisher Scientific Vitrobot Mark IV set to 90% humidity at 10 °C (blotting force, −5; blotting time, 3–3.5 s) or a Leica EM GP2 with sensor-assisted back-side (blotting time, 3–5 s) with the same temperature and humidity settings. Movies were recorded in underfocus on a 200 kV Talos Arctica G2 (Thermo Fisher Scientific) electron microscope equipped with a Bioquantum K3 (Gatan) detector operated by EPU (Thermo Fisher Scientific).

## Image processing and helical reconstruction

Movie frames were gain-corrected, dose-weighted and aligned using cryoSPARC Live (v.3.3.2)[52]. Initial two-dimensional (2D) classes were produced using the auto picker implemented in cryoSPARC Live. The following image processing steps were performed using cryoSPARC. The best-looking classes were used as templates for the filament trace. The resulting filament segments were extracted with 600 pixel box size (Fourier cropped to 200 pixels) and subjected to multiple rounds of 2D classification. The remaining segments were re-extracted with a box size of 400 pixels (Fourier cropped to 200 pixels) and subjected to an additional round of 2D classification. The resulting 2D class averages were used to determine filament diameters and initial symmetry guesses in PyHI (v.8a98c25)[53]. Symmetry guesses were validated by initial helical refinement in cryoSPARC and selection of the helical symmetry parameters yielding reconstructions with typical PspA features and the best resolution. Then all segments were classified by heterogeneous refinement and subsequent 3D classifications using the initial helical reconstructions as templates. The resulting class distribution gave the PspA rod diameter distribution shown in Fig. 2 and Extended Data Fig. 2. The resulting helical reconstructions were subjected to multiple rounds of helical refinement including the symmetry search option. For the final polishing, the segments were re-extracted at 400 pixels without Fourier cropping. Bad segments were discarded by heterogeneous refinement. Higher-order aberrations were corrected by global and local CTF refinement followed by a final helical refinement step. The local resolution distribution and local filtering were performed using cryoSPARC. The resolution of the final reconstructions was determined by Fourier shell correlation (auto-masked, FSC = 0.143). The peaks in some FSC curves at 1/5.3 Å and lower resolution (Extended Data Figs. 2 and 6) correspond to the features (α-helix pitch) of all α-helical structures present in the PspA protein.

## Cryo-EM map interpretation and model building

The 3D reconstructions were B-factor sharpened in Phenix (phenix. auto-sharpen)[54]. The handedness of the final map was determined by rigid-body fitting the PspA reference structure amino acids 22–217 (PDB 7ABK)[9] into the final maps using ChimeraX (v.1.5.dev202206170050)[55,56] and flipped accordingly. The PDB 7ABK structure was fitted using molecular dynamics flexible fitting to the 3D reconstructions in ISOLDE (v.1.7.1)[57]. Then the respective helical symmetry was applied to all models to create assemblies of 60 monomers each using Chimera (v.1.16)[58]. The assembly models were subjected to auto-refinement with phenix.real_space_refine[59] (with NCS constraints and NCS refinement). After auto-refinement, the new models were used for local model-based map sharpening with LocSCALE (v.0.1)[60] to produce the final maps. The auto-refined models were checked and adjusted manually in Coot (v.0.98)[61] and ISOLDE[57] before a final cycle of auto-refinement with phenix.real_space_refine[59] (with NCS constraints and NCS refinement). After the final inspection, the model was validated in phenix.validation_cryoem[62]/Molprobity[63]. Image processing, helical reconstruction and model building were completed using SBGrid-supported applications[64]. In this manner, from a total of 27 cryo-EM maps determined in three samples (apo, ADP, ATP), a total of unique 11 PDB coordinates were refined and deposited originating from the ATP sample (see Table 1). The remaining 16 maps were of identical symmetry consisting of either similar or poorer resolution densities and, therefore, atomic model refinement was not pursued further.

Diameters of the final PspA rod reconstructions were measured using ImageJ[51]. First, radial intensity profiles of each reconstruction were created. Then the radius at an intensity cutoff of 0.3 was read. The radius readout from reconstructions from the same helical symmetry class was averaged, converted to diameters and rounded to 5 Å increments. Outer and inner leaflet radii of engulfed membrane tubes were determined from the same radial intensity profiles at the peak maxima from each bilayer leaflet.

## Analysis of membrane images

For a description of the membrane morphology, subsets of 50–100 micrographs from datasets of sole SUVs (control) or SUVs with PspA or PspA + ATP, respectively, were analyzed. Statistical analysis of the membrane features was performed on segmentations of the micrographs. Multiple micrographs from each dataset were manually segmented at a pixel size of 7 Å and given as patches as a training dataset to a standard U-Net[65] with a depth of 4, patch size of 256, kernel size of 3 and batch size of 32. The micrographs were divided into patches, normalized to a mean of 0 and a s.d. of 1, and simple rotations (90°, 180° and 270°) as well as flipped patches were added as data augmentation. The individual membranes are represented as the skeleton of their segmentation[66], and highly aggregated or overlapping sections of the images were discarded, as the automatic identification of the membrane shapes was ambiguous. Furthermore, only closed-segmented vesicles were analyzed. The perimeter of a vesicle was estimated by calculating the sum of the distances between neighboring points of the skeleton. For automatic analysis of the enclosure distance, the skeleton points of a vesicle were compared with the points of other vesicles in the same micrograph. When the points were found within the skeleton of another vesicle, it was labeled as an enclosed vesicle. A vesicle can be enclosed by and enclose multiple other vesicles. When a vesicle was labeled as enclosed, the minimum Euclidean distance between the two vesicles was calculated. The thickness estimation of membranes was modified from a previous publication[67]. In brief, the image intensities are radially averaged along the membrane skeleton. The bilayer thickness for a membrane segment is then estimated by the distance between the minima in the radially averaged intensity profile for all nearby coordinates.

## Statistics and reproducibility

Data and statistical analysis were performed using OriginPro v.2021b (OriginLab). Detailed descriptions of quantifications and statistical analyses (exact values of $n$, dispersion and precision measures used and statistical tests) can be found in the respective figures, figure legends and Methods. Unless mentioned otherwise, a two-sided $t$-test with two samples was used for all statistical analyses. Shown exemplary micrographs are part of cryo-EM datasets from single samples; sample details and dataset sizes are listed in Extended Data Table 2.

## Reporting summary

Further information on research design is available in the Nature Portfolio Reporting Summary linked to this article.

## Data availability

All unique and stable reagents generated in this study are available from the lead contact with a completed Material Transfer Agreement. Underlying source data are available with the manuscript online. The Electron Microscopy Data Bank (EMDB) accession numbers for cryo-EM maps and PspA models are 15489, 15490, 15491, 15492, 15493, 15494, 15495, 15497, 15496, 15498 and 15499; Protein Data Bank (PDB) IDs are 8AKQ, 8AKR, 8AKS, 8AKT, 8AKU, 8AKV, 8AKW, 8AKY, 8AKX, 8AKZ and 8AL0. Available datasets accessed in this article are PDB 7ABK and EMDB-11698. Source data are provided with this paper.

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

## Acknowledgements

This study was funded by the Deutsche Forschungsgemeinschaft (German Research Foundation; SA 1882/6-1 (C.S.), SCHN 690/16-1 (D.S.), CRC1208 Project No. 267205415 (C.S.) and CRC1551 Project No. 464588647 (D.S.)). We gratefully acknowledge the electron microscopy access time and computing time granted by the biological EM facility of the Ernst-Ruska Centre at Forschungszentrum Jülich. In this regard, we thank T. Heidler and P. Sundermeyer for maintaining the electron microscopes and D. Mann for maintaining the processing computers. We gratefully acknowledge the computing time granted through the Jülich Aachen Research Alliance on the supercomputer JURECA at Forschungszentrum Jülich[49].

## Author contributions

B.J., D.S. and C.S. designed the research. E.H., I.R. and A.H. cloned, expressed and purified the proteins. B.J. and E.H. prepared cryo-EM samples. B.J. and E.H. operated the electron microscopes. B.J., E.H. and C.S. determined the cryo-EM structures. B.J. and E.H. built the refined atomic model. B.J., E.H. and A.H. measured and B.J., E.H., A.H. and D.S. interpreted the ATPase activity data. B.J. and P.S. performed micrograph-based membrane morphology analysis. B.S.S. performed while B.S.S. and P.H. interpreted the mass spectrometry experiments. B.J., E.H., D.S. and C.S. prepared the manuscript with input from all authors.

## Funding

## Competing interests

The authors declare no competing interests.

## Additional information

**Extended data** is available for this paper at https://doi.org/10.1038/s41594-024-01359-7.

**Correspondence and requests for materials** should be addressed to Carsten Sachse.

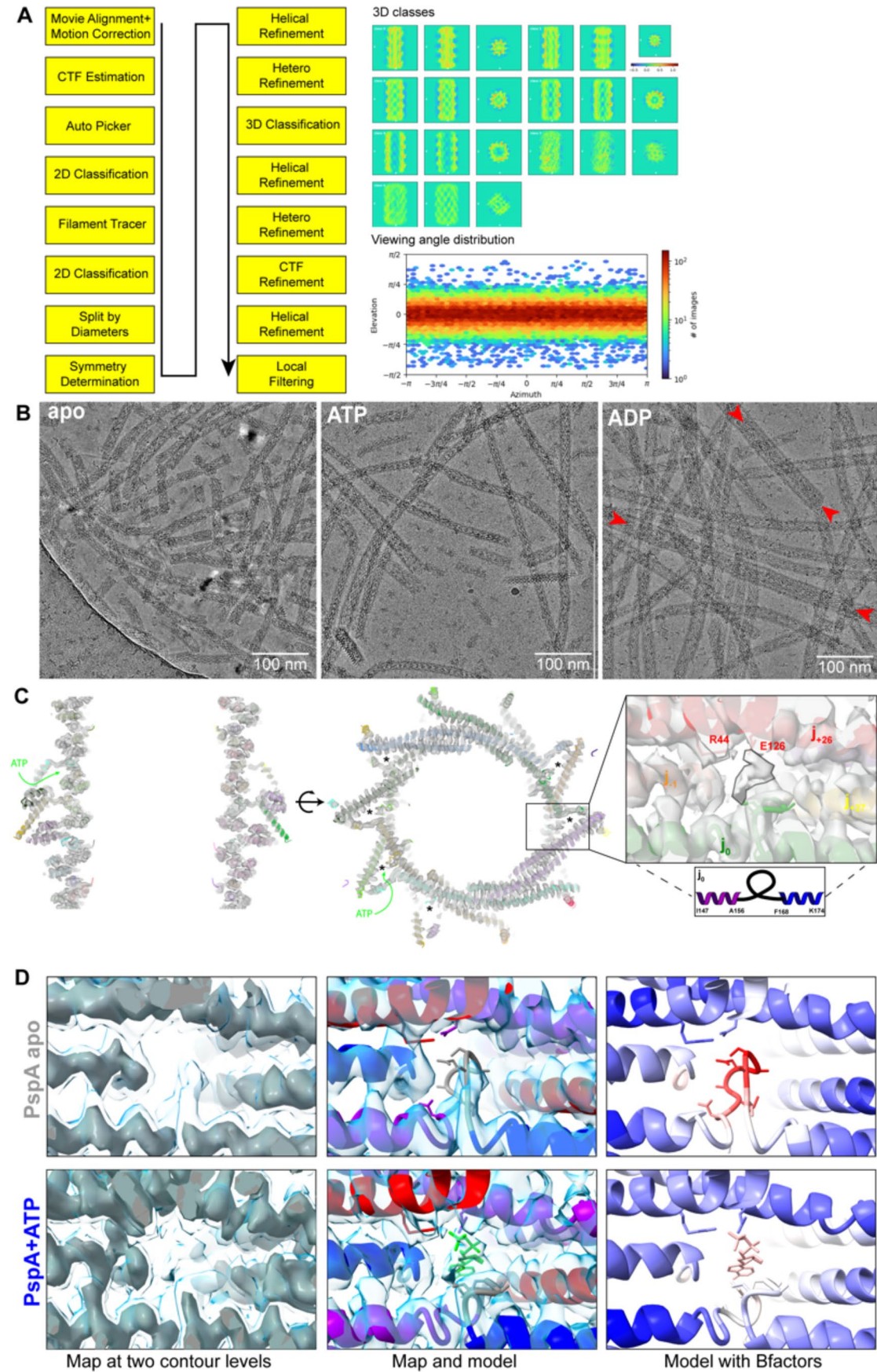

**Extended Data Fig. 1 | See next page for caption.**

**Extended Data Fig. 1 | Cryo-EM Data and Processing. A:** Cryo-EM Processing Workflow, 3D classes and exemplary viewing angle distribution. **B:** Three selected cryo-EM micrographs of the three corresponding PspA data sets containing 1818, 4470, and 3038 movies. *Apo*: PspA without additives. ATP: PspA with 2 mM ATP. ADP: PspA with 2 mM ADP; red arrows show "super rods" with engulfed "normal" rods. **C:** Sliced side and top view of the PspA cryo-EM density in the presence of ATP (200 Å diameter, PDB: 8AKR, EMDB: 15490) with a superimposed atomic PspA model. Putative nucleotide-binding sites are labeled with asterisks. Green arrows indicate solvent accessibility of the ATP binding site. Inset: Zoomed view of the putative nucleotide-binding site in the PspA rod structure (200 Å diameter) with weak segmented density above the loop connecting helices α3 and α4 (cyan) due to incomplete binding and imposition of helical symmetry.

The nucleotide-binding site is formed by the surrounding chains (subunits $j_0$, $j_{-1}$, $j_{+26}$, and $j_{+27}$) and is repeated for every subunit in the polymer. Inset. View of the cryo-EM density of the nucleotide binding region superimposed with PspA, no nucleotide model included in density (black rim). **D:** Comparison of the PspA apo structure with PspA after incubation with ATP. Left column: Maps of PspA and PspA+ATP at low and high contour levels superimposed. Middle column: Models of PspA apo (PDB:7ABK) and PspA+ATP with an ADP model fitted to the additional density. Note that due to the presence of the nucleotide the loop in the PspA+ATP structure is moved. Right column: Atomic models of PspA apo and PspA+ATP superimposed with colored atomic temperature factors indicate increased mobility of α3/α4 loop in the PspA apo structure.

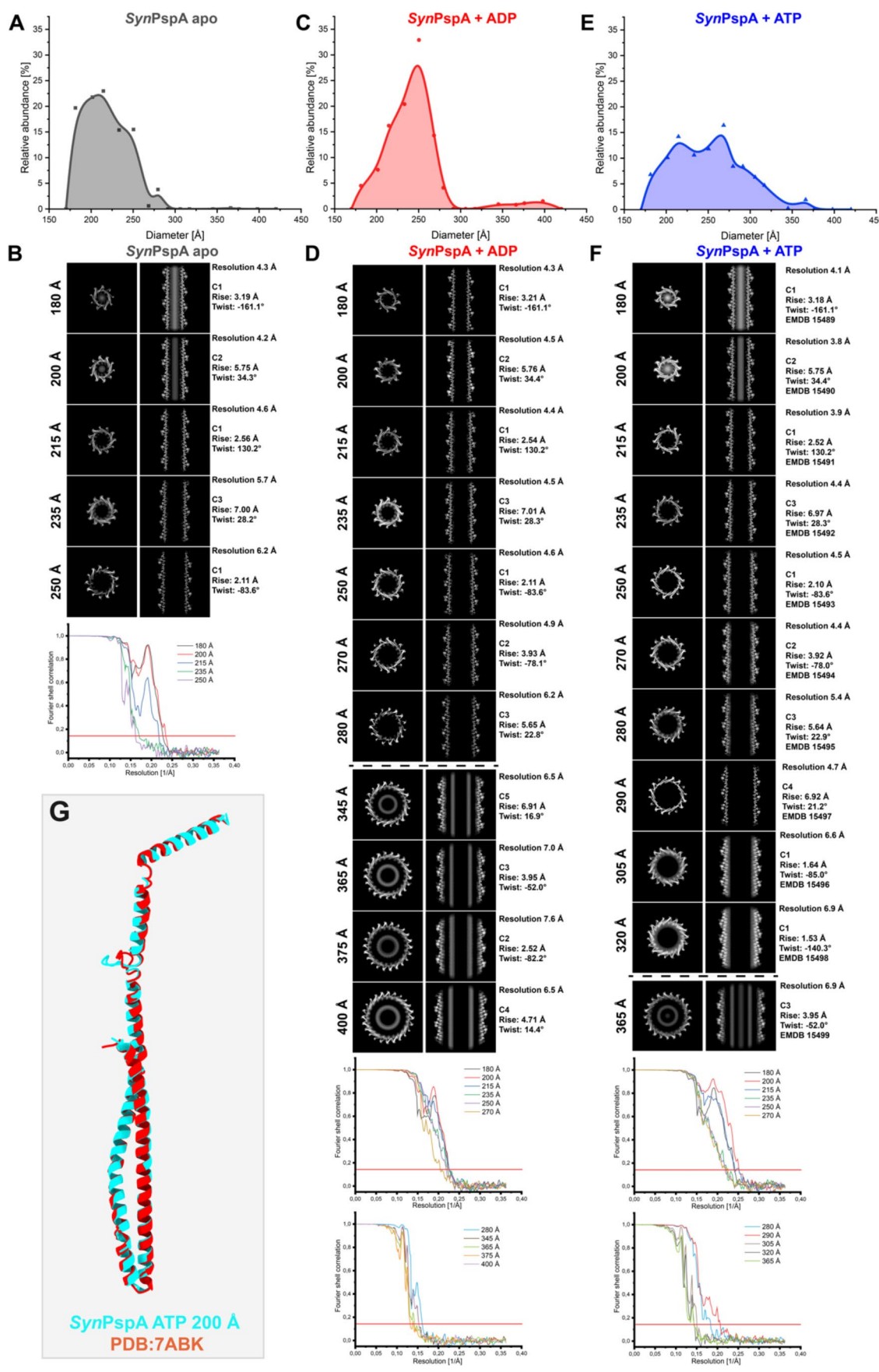

**Extended Data Fig. 2 | See next page for caption.**

**Extended Data Fig. 2 | PspA rod diameter distributions and corresponding cryo-EM structures in the absence or presence of nucleotides.** PspA rod diameter distribution for **A:** PspA *apo* (grey), **C:** PspA+ADP (red), and **E:** PspA+ATP (blue), based on the relative occurrence of rod segments with a certain diameter. **B, D, F:** Overview of PspA rod cryo-EM structures with cross-sectional top view *z*-slices (left column), cross-sectional side view *xy*-slices (middle column) and FSC curves with a 0.143 threshold of **B:** PspA *apo*, **D:** PspA+ADP and **F:** PspA+ATP. Note that when FSC curves drop below 1, systematic peaks of high correlation occur at 1/5.3 Å corresponding to the α-helical pitch feature and additional low-resolution details of the all α-helical maps of PspA. **G:** Comparison of the PspA ATP 200 Å monomer with the high-resolution reference structure of PspA *apo* PDB:7ABK.

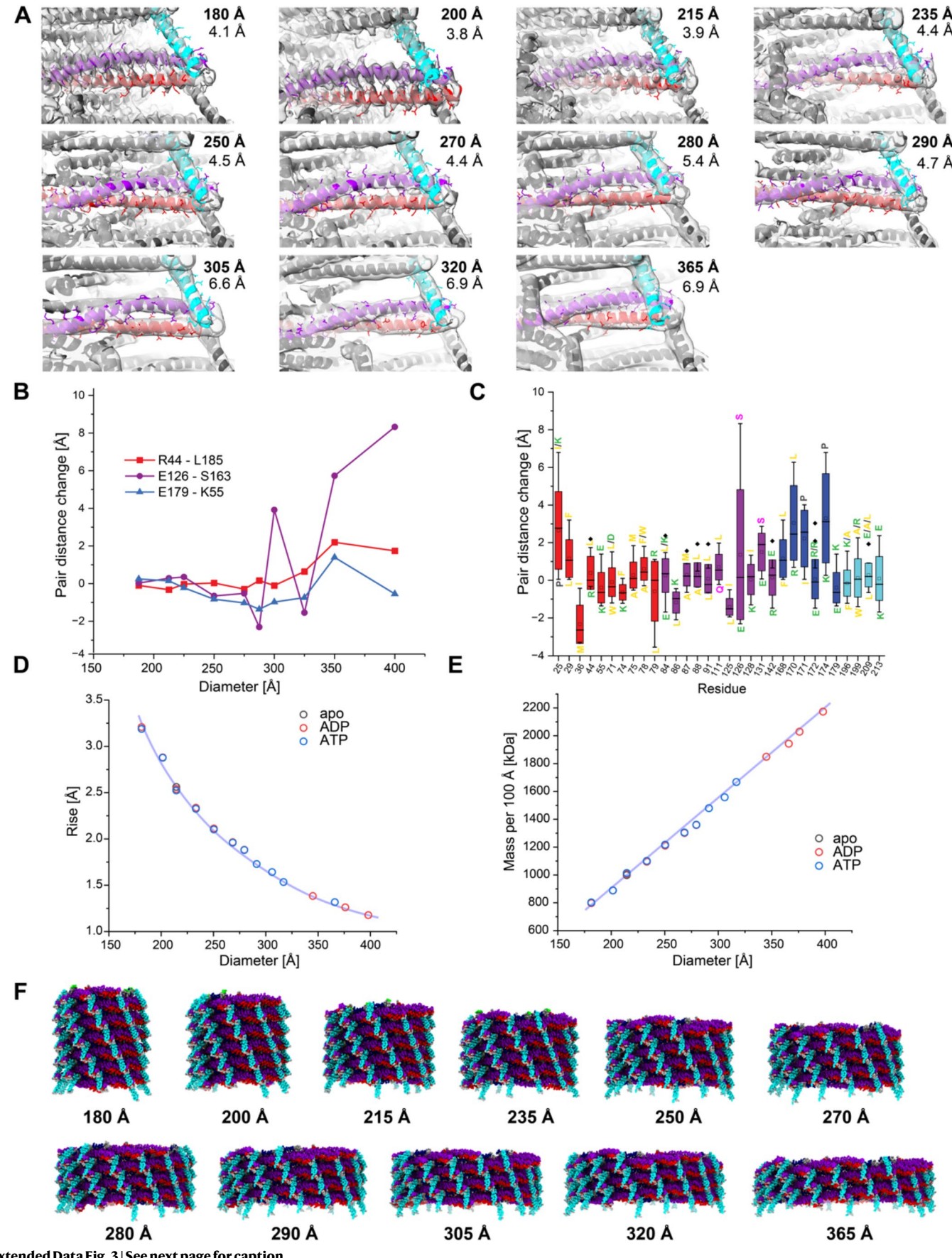

**Extended Data Fig. 3 | See next page for caption.**

**Extended Data Fig. 3 | PspA rod radial density profiles and mass per length.**
**A:** Cryo-EM density fit of atomic PspA models of different diameters focusing on the helices α1/α2 hairpin and helix 5 (α1 red, α2 + 3 violet, α4 blue, α5 cyan). (180 Å: PDB 8AKQ, 200 Å: PDB 8AKR, 215 Å: PDB 8AKS, 235 Å: PDB 8AKT, 250 Å: PDB 8AKU, 270 Å: PDB 8AKV, 280 Å: PDB 8AKW, 290 Å: PDB 8AKY, 305 Å: PDB 8AKX, 320 Å: PDB 8AKZ, 365 Å: PDB 8AL0) **B:** Scatter plot of three selected pairs (R44-L185, E126-S163, E179-K55) distance changes with respect to the initial distance in the 180 Å diameter assembly over all rod diameters. **C:** Box plot of pair distance changes of evolutionarily conserved residues with respect to the initial distance in the 180 Å diameter assembly. The residues were selected by first identifying potential intermolecular interactions between highly conserved residues ( > 90% conserved among PspA/Vipp1 proteins[9]. Then, the Cα distance for each pair was measured for each rod diameter. To calculate the difference of the pair distances relative to the smallest diameter rods, the distances in 180 Å

rods were subtracted from the respective distances in the other diameters. The s.d. of the distance shift distribution is a measure of the flexibility of the interaction. Boxes show SD with median line (line) and mean value (circle). Whiskers show the range within 1.5 IQR. Color code for residues: green=charged; yellow=hydrophobic; pink=polar+uncharged: grey=special cases. Color code for boxes: residues located in helix α1: red, residues located in helix α2 + 3: violet, residues located in helix α4: blue, residues located in helix α5: cyan. n = 11; n: Number of different rod diameters used for distance measurement.
**D**: Diameters plotted against helical rise (after correction for number of strands) in Å. *Apo* (black): PspA ADP (red): PspA + 2 mM ADP. ATP (blue): *PspA* + 2 mM ATP. **E:** Diameters plotted against mass per 100 Å length in kDa. *Apo* (black): *PspA*. ADP (red): *PspA* + 2 mM ADP. ATP (blue): PspA + 2 mM ATP. **F:** PspA models with 60 monomers each show a decrease in rod length and an increasing diameter from 180 - 365 Å.

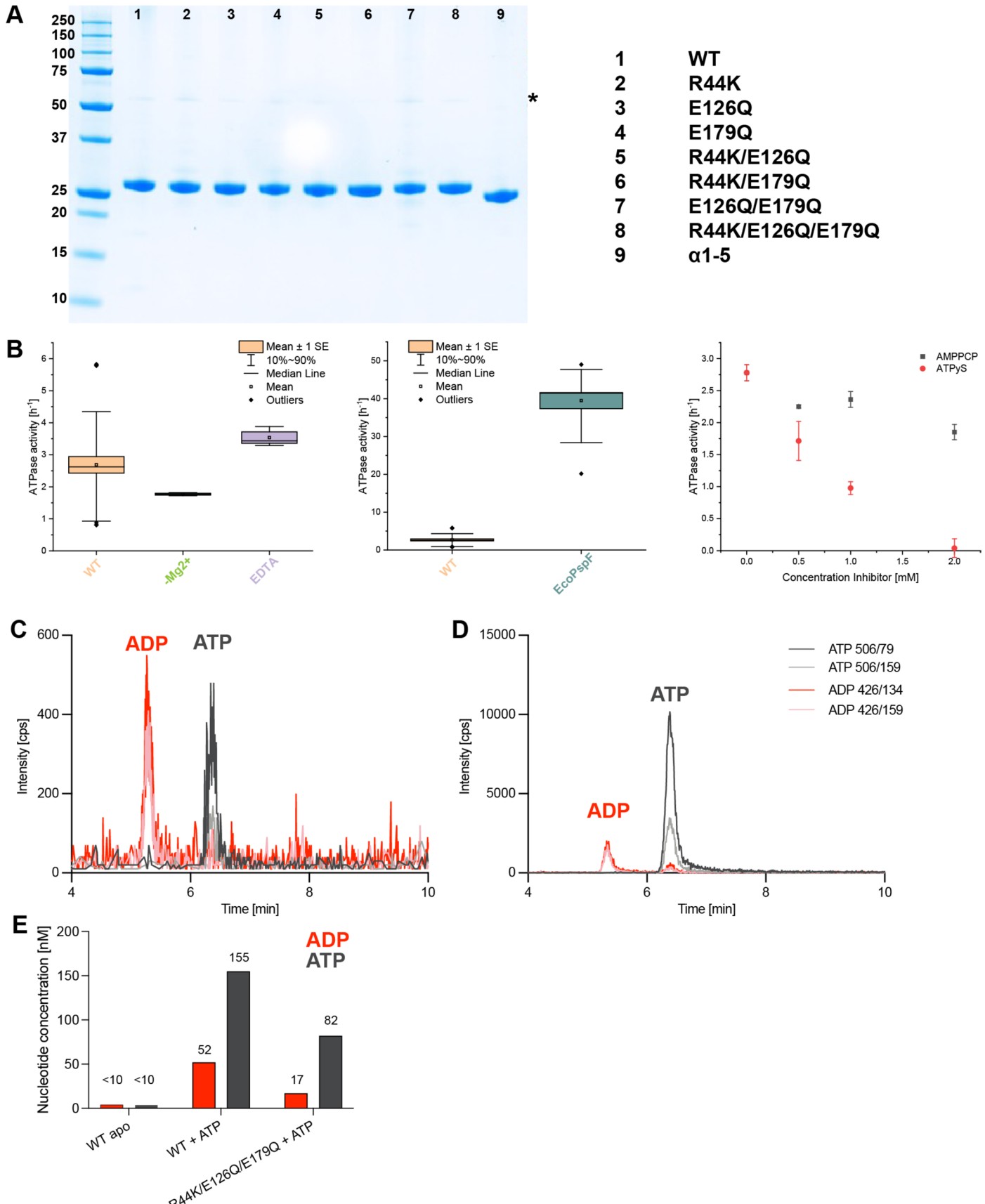

**Extended Data Fig. 4 | See next page for caption.**

**Extended Data Fig. 4 | Nucleotide binding and hydrolysis by PspA wild-type (WT) and mutants. A:** SDS PAGE of purified PspA WT and mutants (3 µg each). Purified proteins show a band at 28 kDa, except for the α0 truncated form which shows a band at 25 kDa. Asterisk: Faint dimer bands at twice the height of the monomeric protein band (56 kDa for WT and full-length PspA mutants; 50 kDa for α1-5 mutant). Marker: Bio-Rad Precision PlusProteinTM Unstained standards. **B:** ATPase activity of PspA. Left graph: Influence of $Mg^{2+}$ and EDTA on the ATPase activity of the WT protein. WT (orange): ATPase activity in the presence of 2 mM $Mg^{2+}$. -$Mg^{2+}$(green): ATPase activity in the absence of $Mg^{2+}$. EDTA (purple): ATPase activity in the presence of 10 mM EDTA; n(WT) = 36, n(-$Mg^{2+}$ and EDTA) = 3. The boxplots show the mean ± standard error as boxes, the 10% to 90% percentile as whiskers, and outliers as diamonds. Middle graph: Comparison of the PspA ATPase activity with ATPase activity of *E. coli* PspF, n(PspA)=36, n(*Eco*PspF)=3.

The boxplots show the mean ± standard error as boxes, the 10% to 90% percentile as whiskers, and outliers as diamonds. Right graph: Influence of ATPyS and AMPPCP on the ATPase activity of PspA. Red circle: ATPyS; Grey square: AMPPCP; n = 3, error bars represent SD. **C:** HPLC/MS-MS of PspA directly after purification. **D:** HPLC/MS-MS of PspA R44K/E126Q/E179Q + ATP after 24 h incubation and extensive washing. Different color lines represent different MRM transitions. For ADP, MRM transitions are 426/134 (red) and 426/159 (light red). For ATP MRM transitions are 506/79 (dark grey) and 506/159 (light grey). The ADP fragments below the ATP peak are formed by in-source fragmentation of ATP in the ESI source. **E:** Bar plot of estimated nucleotide concentrations found by LC-MS/MS in PspA *apo* control, WT, and R44K/E126Q/E179Q rods after incubation with ATP for 24 h and extensive washing, n = 1.

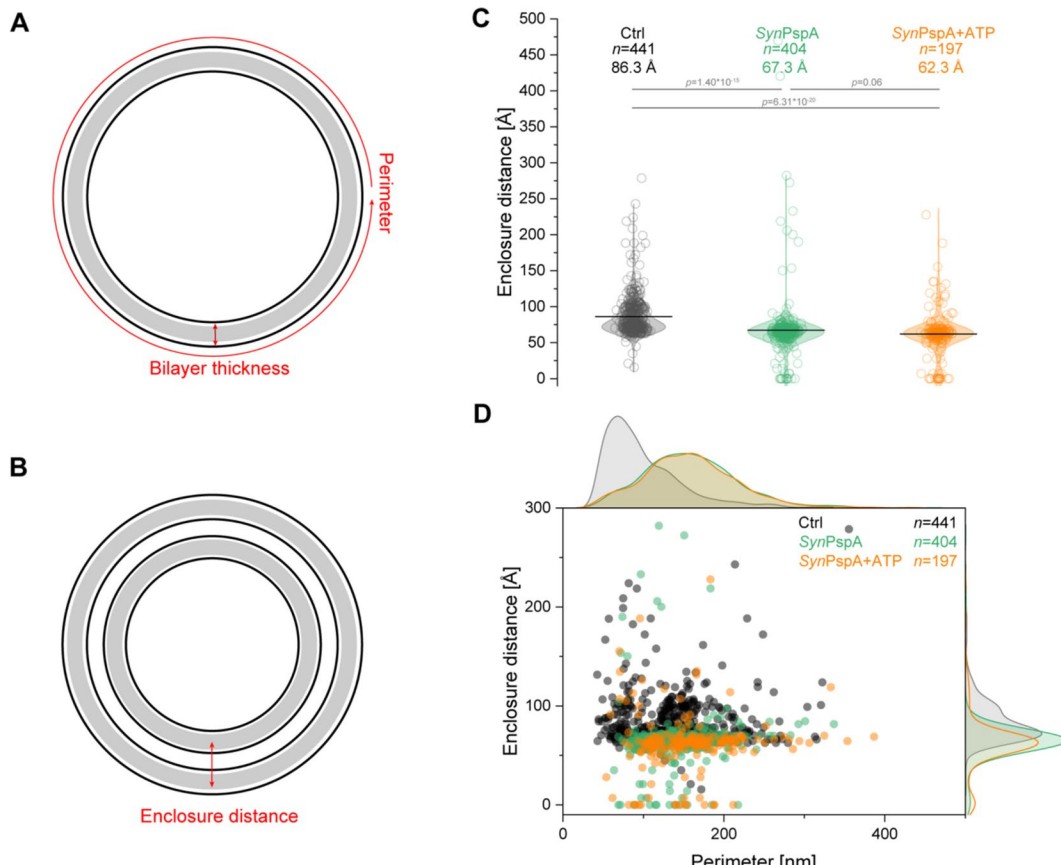

**Extended Data Fig. 5 | Morphological changes of EPL vesicles after PspA/ATP reconstitution. A + B:** Schematic view of the measured vesicle parameters: perimeter, bilayer thickness, and enclosure distance (see materials and methods for details). **C:** Violin plot of the enclosure distance of control EPL SUVs (grey), PspA+EPL SUVs (green), and PspA+EPL + ATP SUVs (orange). The mean enclosure distance, and the number of measured vesicles *n* and *p* values from a two-sample t-test are indicated on the graph. **D:** Scatter plot and histogram of the vesicle perimeter versus enclosure distance of control EPL SUVs (grey), PspA+EPL (green), and PspA+EPL + ATP (orange). The number of measured vesicles *n* is indicated on the graph.

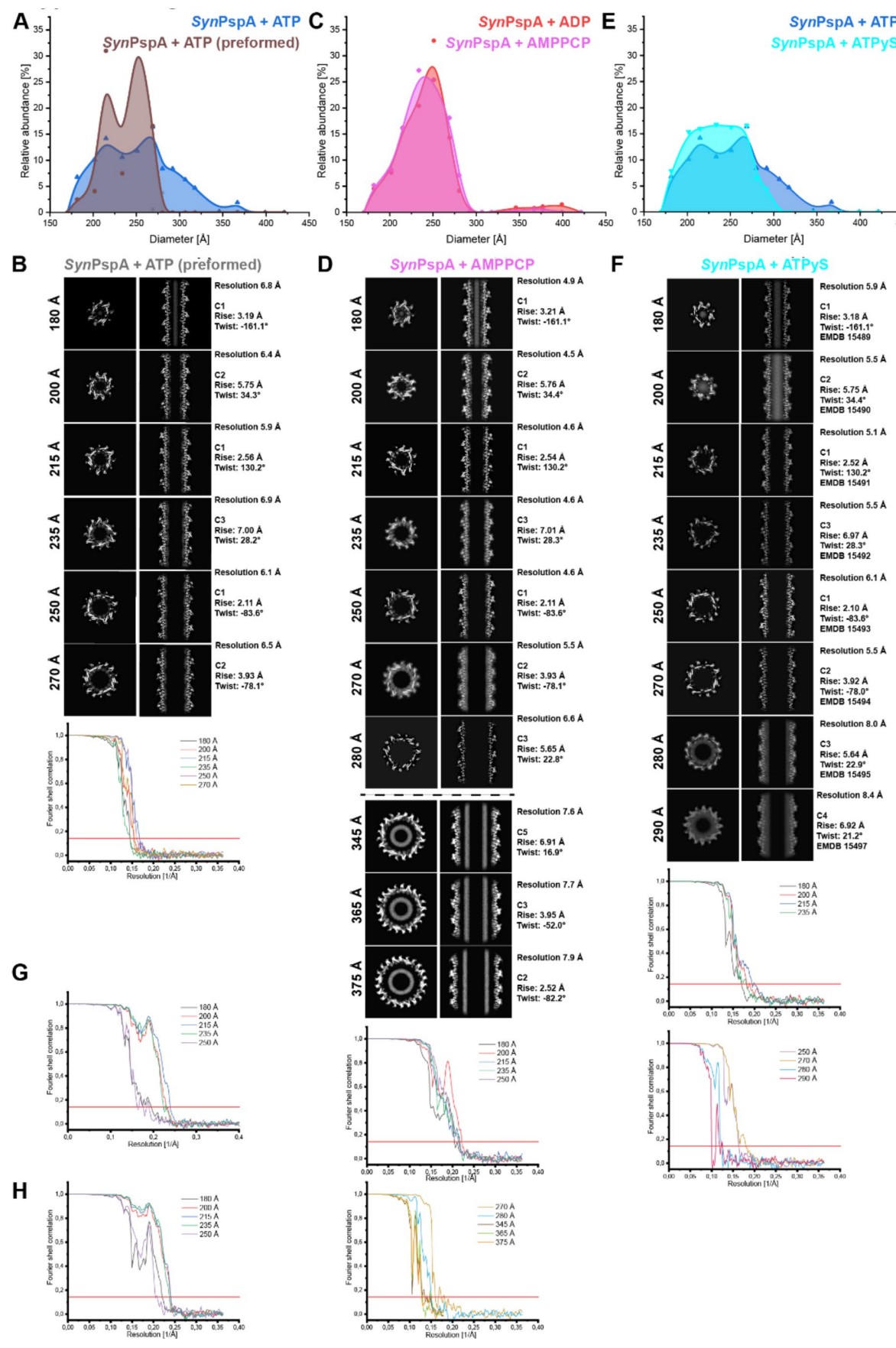

**Extended Data Fig. 6 | See next page for caption.**

**Extended Data Fig. 6 | PspA rod diameter distribution and corresponding cryo-EM structures in the presence of non-hydrolyzable ATP analogs. Top**. PspA rod diameter distribution for **A:** PspA+ATP (blue), and preformed PspA rods incubated with ATP (brown), **C:** PspA+ADP (red) and PspA+AMPPCP (magenta), and **E:** PspA+ATP (blue) and PspA+ATPγS (cyan), based on the relative occurrence of rod segments with a certain diameter. **Bottom**. Cross-sectional top view *z*-slices (left column), cross-sectional side view *xy*-slices (middle column) and FSC curves (threshold 0.143) of cryo-EM structures **B:** PspA+ATP (preformed), **D:** PspA+AMPPCP and **F:** PspA+ATPγS. **G:** FSC curve of PspA dL10 diameters (threshold 0.143). **H:** FSC curve of PspA dL10 diameters with ATP (threshold 0.143). Note that when FSC curves drop below 1, systematic peaks of high correlation occur at 1/5.3 Å corresponding to the α-helical pitch and additional low-resolution features of the all α-helical maps of PspA.

**Extended Data Table 1 | Parameters for the quantification of ATP/ADP**

| Compound | Precursor (*m/z*) | Product ion (*m/z*) | DP (eV) | CE (eV) | CXP (eV) | Retention time (min) |
|---|---|---|---|---|---|---|
| ATP | 506 | 79 (quantifier) | -105 | -30 | -11 | 6.3 |
| | | 159 (qualifier) | -105 | -34 | -9 | |
| ADP | 426 | 134 (quantifier) | -120 | -39 | -7 | 5.2 |
| | | 159 (qualifier) | -120 | -34 | -9 | |

**Extended Data Table 2 | Sample Details**

| | SynPspA apo | SynPspA ATP | SynPspA ADP | SynPspA AMPPCP | SynPspA ATPγS | SynPspA dL10 | SynPspA dL10 ATP | SynPspA ATP preformed | SynPspA EPL | SynPspA EPL ATP | SynPspA R44K/E126Q/E179Q/ EPL ATP | EPL |
|---|---|---|---|---|---|---|---|---|---|---|---|---|
| Protein Conc. | 6 − 8 mg/mL | 6 − 8 mg/mL | 6 − 8 mg/mL | 6 − 8 mg/mL | 6 − 8 mg/mL | 0.5 mg/mL | 0.5 mg/mL | 0.1 mg/mL | 1.2 mg/mL | 1.2 mg/mL | 0.9 mg/mL | - |
| Lipid Conc. | - | - | - | - | - | - | - | - | 2.5 mg/mL | 2.5 mg/mL | 2.5 mg/mL | 2.5 mg/mL |
| NTP Conc. | - | 2 mM | 2 mM | 1 mM | 0.5 | - | 2 mM | 2 mM | - | 10 mM | 10 mM | - |
| MgCl$_2$ Conc. | - | 4 mM | 4 mM | 2 mM | 4 mM | - | 4 mM | 4 mM | - | 20 mM | 20 mM | - |
| Magnification | 63 kx | 63 kx | 63 kx | 63 kx | 63 kx | 105 kx | 63 kx | 63 kx | 63 kx | 63 kx | 63 kx | 63 kx |
| Pixel size | 1.362 Å | 1.362 Å | 1.362 Å | 1.362 Å | 1.362 Å | 0.84 | 1.362 Å | 1.362 Å | 1.362 Å | 1.362 Å | 1.362 Å | 1.362 Å |
| Frames | 40 | 40 | 40 | 40 | 40 | 35 | 45 | 45 | 40 | 40 | 40 | 40 |
| Total dose | 58 e−/Å2 | 58 e−/Å2 | 58 e−/Å2 | 58 e−/Å2 | 58 e−/Å2 | 35 e−/Å2 | 45 e−/Å2 | 45 e−/Å2 | 58 e−/Å2 | 58 e−/Å2 | 58 e−/Å2 | 58 e−/Å2 |
| Defocus range | 1.0 to -3.0 µm | 1.0 to 4.0 µm | 1.5 to 3.5 µm | 1.5 to 3.5 µm | 1.0 to 4.0 µm | 1.5 to 3.0 µm | 1.5 to 3.0 µm | 1.5 to 3.0 µm | 1.5 to 3.5 µm | 2.0 to 4.0 µm | 2.0 to 4.0 µm | 2.0 to 4.0 µm |
| Movies | 1,818 | 4,470 | 3,038 | 3,127 | 3,310 | 3,620 | 4,311 | 3,065 | 50 | 61 | 73 | 632 |

# Reporting Summary

## Statistics

For all statistical analyses, confirm that the following items are present in the figure legend, table legend, main text, or Methods section.

| n/a | Confirmed | |
|---|---|---|
| ☐ | ☒ | The exact sample size (*n*) for each experimental group/condition, given as a discrete number and unit of measurement |
| ☐ | ☒ | A statement on whether measurements were taken from distinct samples or whether the same sample was measured repeatedly |
| ☐ | ☒ | The statistical test(s) used AND whether they are one- or two-sided *Only common tests should be described solely by name; describe more complex techniques in the Methods section.* |
| ☒ | ☐ | A description of all covariates tested |
| ☒ | ☐ | A description of any assumptions or corrections, such as tests of normality and adjustment for multiple comparisons |
| ☐ | ☒ | A full description of the statistical parameters including central tendency (e.g. means) or other basic estimates (e.g. regression coefficient) AND variation (e.g. standard deviation) or associated estimates of uncertainty (e.g. confidence intervals) |
| ☐ | ☒ | For null hypothesis testing, the test statistic (e.g. *F*, *t*, *r*) with confidence intervals, effect sizes, degrees of freedom and *P* value noted *Give P values as exact values whenever suitable.* |
| ☒ | ☐ | For Bayesian analysis, information on the choice of priors and Markov chain Monte Carlo settings |
| ☒ | ☐ | For hierarchical and complex designs, identification of the appropriate level for tests and full reporting of outcomes |
| ☒ | ☐ | Estimates of effect sizes (e.g. Cohen's *d*, Pearson's *r*), indicating how they were calculated |

*Our web collection on statistics for biologists contains articles on many of the points above.*

## Software and code

Policy information about availability of computer code

| Data collection | EPU (v2.12.1.2782) |
|---|---|
| Data analysis | CryoSPARC (v3.3.2), Chimera (v1.16), ChimeraX (v1.5.dev202206170050), Coot (v0.98), Phenix (v1.20.1-4487), LocScale 0.1, ImageJ/Fiji 2.14, PyHI 8a98c25, OriginPro 2021b |

For manuscripts utilizing custom algorithms or software that are central to the research but not yet described in published literature, software must be made available to editors and reviewers. We strongly encourage code deposition in a community repository (e.g. GitHub). See the Nature Portfolio guidelines for submitting code & software for further information.

## Data

Policy information about availability of data

All manuscripts must include a data availability statement. This statement should provide the following information, where applicable:
- Accession codes, unique identifiers, or web links for publicly available datasets
- A description of any restrictions on data availability
- For clinical datasets or third party data, please ensure that the statement adheres to our policy

The EMDB accession numbers for cryo-EM maps and PspA models depostided with this Manuscript are EMD IDs: 15489, 15490, 15491, 15492, 15493, 15494, 15495, 15497, 15496, 15498, 15499

and PDB-IDs: 8AKQ, 8AKR, 8AKS, 8AKT, 8AKU, 8AKV, 8AKW, 8AKY, 8AKX, 8AKZ, 8AL0.
Available datasets accessed in this Manuscript are: PDB-ID: 7ABK, EMD ID: 11698

# Research involving human participants, their data, or biological material

Policy information about studies with human participants or human data. See also policy information about sex, gender (identity/presentation), and sexual orientation and race, ethnicity and racism.

| | |
|---|---|
| Reporting on sex and gender | N.A. |
| Reporting on race, ethnicity, or other socially relevant groupings | N.A. |
| Population characteristics | N.A. |
| Recruitment | N.A. |
| Ethics oversight | N.A. |

Note that full information on the approval of the study protocol must also be provided in the manuscript.

# Field-specific reporting

Please select the one below that is the best fit for your research. If you are not sure, read the appropriate sections before making your selection.

☒ Life sciences   ☐ Behavioural & social sciences   ☐ Ecological, evolutionary & environmental sciences

For a reference copy of the document with all sections, see nature.com/documents/nr-reporting-summary-flat.pdf

# Life sciences study design

All studies must disclose on these points even when the disclosure is negative.

| | |
|---|---|
| Sample size | A total of 1818, 4470, 3038, 3127, 3310, 3620, 4311, 3065, 50, 61, 73, and 632 movies were collected for PspA apo, PspA ATP, PspA ADP, PspA AMPPCP, PspA ATPgS, PspA dL10, PspA ATP preformed, PspA EPL, PspA EPL ATP, R44/E126Q/E179Q/EPL ATP and EPL samples, respectively. Sample sizes were chosen based on instrument access time. The number of micrographs was sufficient to produce high resultion reconstructions of the samples. |
| Data exclusions | Micrographs of poor particle coverage and ice quality were discarded. |
| Replication | Due to the time-consuming nature of image acquisition and the limited access to this specialized microscope equipment (high-end Krios and Arctica microscopes), exact replicates were not performed. |
| Randomization | Randomization is not applicable to the study because of the time-consuming nature of image acquisition and the limited access to this specialized microscope equipment (high-end Krios and Arctica microscopes). |
| Blinding | Blinding experiments is not applicable to this study because of time-consuming nature of image acquisition and the limited access to this specialized microscope equipment (high-end Krios and Arctica microscopes). |

# Reporting for specific materials, systems and methods

We require information from authors about some types of materials, experimental systems and methods used in many studies. Here, indicate whether each material, system or method listed is relevant to your study. If you are not sure if a list item applies to your research, read the appropriate section before selecting a response.

## Materials & experimental systems

| n/a | Involved in the study |
|---|---|
| ☒ | ☐ Antibodies |
| ☒ | ☐ Eukaryotic cell lines |
| ☒ | ☐ Palaeontology and archaeology |
| ☒ | ☐ Animals and other organisms |
| ☒ | ☐ Clinical data |
| ☒ | ☐ Dual use research of concern |
| ☒ | ☐ Plants |

## Methods

| n/a | Involved in the study |
|---|---|
| ☒ | ☐ ChIP-seq |
| ☒ | ☐ Flow cytometry |
| ☒ | ☐ MRI-based neuroimaging |

