## [Peer Review File · Nature Structural & Molecular Biology]

Peer Review Information

Manuscript Title: Structural plasticity of bacterial ESCRT-III protein PspA in higher-order assemblies

Corresponding author name(s): Carsten Sachse

Editorial Notes:

Reviewer Comments & Decisions:

Decision Letter, initial version:

Message:

Nature Structural & Molecular Biology NSMB-A47957

6th Sep 2023

Dear Dr. Sachse,

Thank you for submitting your manuscript, "Structural plasticity of bacterial ESCRT-III protein PspA in higher order assemblies". The comments of 2 expert referees are below. You will see that the reviewers have serious concerns and do not find the paper appropriate for publication in our journal. Specifically, reviewer #2 notes conceptual novelty concerns, considering the known literature on other ESCRTIII proteins such as CHMP1B. Both reviewers bring up the fact that the angle of ATP hydrolysis by PspA is underdeveloped. Based on these comments, we cannot offer to publish the study in Nature Structural & Molecular Biology. We hope the referees' comments will be useful to you in revising the manuscript for submission elsewhere.

Given the strengths of the work, we took the liberty to consult our colleagues at Nature Communications. The editors at Nature Communications are interested in the work and, depending on the extent and nature of your revisions, will send the suitably revised manuscript back to reviewers or assess the manuscript editorially in house. In particular, your revision should place the findings in the context of the existing literature, in line with suggestion of referee #2, and discuss the caveats pertaining to the observed binding of ATP, in line with comments of referee #1. If you have any queries, or would like to discuss the extent of revisions necessary, please contact Katarzyna Marcinkiewicz (katarzyna.marcinkiewicz@us.nature.com) who will be the handling editor there. The link to transfer the manuscript is at the end of this email.

I am sorry we could not be more positive on this occasion.

Sincerely,
Kat

Katarzyna Ciazynska
(she/her)
Associate Editor
Nature Structural & Molecular Biology
<https://orcid.org/0000-0002-9899-2428>

Reviewer #1 (Remarks to the Author):

In this manuscript, Junglas and co-workers report the cryo-EM structure of multiple filaments architectures formed by the E. Coli ESCRT-III homologous protein PspA. They also report that PspA possesses ATPase activity, and they have obtained cryo-EM maps of the filaments in the presence of ATP or ADP, showing additional filamentous architectures.

The structural work reported here is very impressive; the authors were able to identify and reconstruct multiple helical symmetries within the data, which is no small feat. The only comment I have for this, is that a couple of additional elements should be included in the supplementary data. Notably, the FSC curves for each map should be added, and perhaps a view of a monomer, with the corresponding density, should be shown as well.

However, I find the ATPase aspect of the story a lot less convincing. Clearly the protein has very low ATPase activity, and the role of ATP binding for the protein function is not at all clear. Critically, the authors report that they do not see any density for the nucleotide in any of the structures that they have obtained. The most compelling evidence is the fact that they observed other filament architectures in the presence of nucleotide; however, the process of grid freezing for cryo-EM is not highly reproducible, and these could merely be caused by slight changes in protein concentration or buffer composition. My concern here is that the authors are reporting non-specific binding of nucleotide, because of a positively-charged pocket, but that this is not related to its function. In the absence of functional data indicating how ATP binding and hydrolysis permits PspA function, this aspect of the manuscript is not entirely compelling.

While I am supportive of publishing this work in NSMB, I would recommend to revise the manuscript, with much less emphasis on the ATPase binding and hydrolysis, and a much more cautious tone about its interpretation.

Reviewer #2 (Remarks to the Author):

In this manuscript, Junglas et al. investigate the molecular determinants of conformational flexibility in the ESCRT-III-family bacterial protein PspA. They use cryoEM to build atomic models for PspA filaments with various diameters and examine how the PspA monomer conformation and inter/intra-molecular interactions change to allow for the assembly of filaments of different diameters. This analysis reveals critical flexibility at hinge regions between alpha helices and specific pairwise interactions that either remain the same or change interacting residues in response to filament diameter changes. Finally, the authors show that, in vitro, PspA filaments display a low level of ATPase activity that may be related to changes in filament diameter. This work complements a body of literature showing that eukaryotic ESCRT-III proteins can assemble into various polymers and copolymers with highly variable curvature. These assemblies and their dynamics are thought to be mechanistically related to the roles of ESCRT proteins in reshaping membranes and catalyzing membrane fission. This manuscript is clearly written and contains numerous cryoEM structures and atomic models.

Major Issues:

- 1) The authors identify the hinge between $\alpha 3$ and $\alpha 4$ as moving the most between

filaments of different diameters, followed by the hinge between $\alpha 2$ and $\alpha 3$. This result is consistent with the movement of the homologous hinge regions for ESCRT-III protein CHMP1B. Nguyen et al. NSMB 2020 (<https://doi.org/10.1038/s41594-020-0404-x>) showed that the very same hinges flex when comparing 180 and 280 Å diameter CHMP1B filaments. This prior work is not mentioned in the current manuscript. It is helpful to see that still further flexing of these same hinges accounts for the formation of still wider PspA filaments and, therefore, that this mechanism is conserved across domains of life in the ESCRT-III family. However, Junglas et al. should frame their results within the context of the published work that established this conceptual precedent.

2) The observation that PspA can hydrolyze ATP *in vitro* is interesting, but more should be done to characterize the role, if any, this activity has in filament structure. For example, ADP/ATP were always added during PspA refolding and assembly. What is the effect on filament structure if the ATP is added after filament assembly? Does adding AMPPCP or ATP γ S during filament formation in the presence of ATP shift the distribution of filament diameters towards smaller diameters?

Minor Issues:

- 1) Line 89, "cyanobacterium" is misspelled.
- 2) In discussing ATP usage by membrane remodeling proteins, the authors may want to consider discussing the case of the dynamin family protein EHD1. Unlike other dynamin family proteins and in analogy to PspA, ATP hydrolysis causes an expansion of the helical filaments (Deo et al. Nat Comm 2018, <https://doi.org/10.1038/s41467-018-07586-z>).

** As a service to authors, Springer Nature Limited provides authors with the ability to transfer a manuscript that one journal cannot offer to publish to another journal, without the author having to upload the manuscript data again. To transfer your manuscript to another NPG journal using this service, please click on [Redacted]

Author Rebuttal to Initial comments

We thank the referees for the positive and constructive feedback on our manuscript NSMB-A47957. According to their comments, we revised the manuscript to highlight the novelty of this comprehensive structural study, to address the gaps in the ATP treatment and to complement the results with functional aspects of membrane remodeling activities. In order to accomplish these tasks, we included a total of 34 newly determined structures, from eight cryo-EM data sets recorded in different conditions, resulting in 4 new composite Figures and additional Table material. Most importantly, our additional experiments revealed new evidence that ATPase activity is linked functionally to membrane remodeling and that we modulated the observed plasticity by a loop mutant that we also show is critical for ATPase activity. We hope that the generated data satisfies the concerns of the reviewers and provides the requested evidence of PspA's structural plasticity.

Reviewer #1 (Remarks to the Author):

In this manuscript, Junglas and co-workers report the cryo-EM structure of multiple filaments architectures formed by the E. Coli ESCRT-III homologous protein PspA. They also report that PspA possesses ATPase activity, and they have obtained cryo-EM maps of the filaments in the presence of ATP or ADP, showing additional filamentous architectures.

1. The structural work reported here is very impressive; the authors were able to identify and reconstruct multiple helical symmetries within the data, which is no small feat. The only comment I have for this, is that a couple of additional elements should be included in the supplementary data. Notably, the FSC curves for each map should be added, and perhaps a view of a monomer, with the corresponding density, should be shown as well.

We appreciate the comment on complementing the data regarding the structural work. Therefore, we now included as requested: Views of the fitted models including the density are shown in Suppl. Fig. 3A. Moreover, we added FSC curves of all shown maps to the respective Suppl. Figures 2 and 6.

2. However, I find the ATPase aspect of the story a lot less convincing. Clearly the protein has very low ATPase activity, and the role of ATP binding for the protein function is not at all clear. Critically, the authors report that they do not see any density for the nucleotide in any of the structures that they have obtained.

We fully share the view with Reviewer #1 that the ATPase activity is low and not comparable to highly active AAA+-ATPases, such as PspF. Therefore, we openly included PspF as a reference. Regardless of this comparison, we clearly showed already in our first manuscript version:

1. HPLC and mass spectrometry demonstrated ATP binding to PspA filaments and turnover to ADP (Suppl. Fig. 4).
2. The enzymatic activity was estimated by two assays: ADP and phosphate release assays (Fig. 4A and Fig. 4B) including replicates and statistically significant figures. Moreover, the activity can be fully shut down by mutating residues in close vicinity to the ATP binding site.

In the revised version of the manuscript, we now added:

Addition 1: Our previous conclusion regarding the lack of density was based on a simple difference map between an apo and an ATP-binding structure, which only led to a small difference density and likely a misleading statement. Given the remark of this reviewer, we reinspected the density and within a limited subset of diameters we able to fit an ADP molecule into a compatible density followed by model refinement. We came to a more differentiated explanation of those densities and now state (page 6, new Suppl. Fig 1B and 1C):

“It is important to note that for rod diameters 180, 200, 215 and 235 Å we identified density in the putative nucleotide binding site close to the loop connecting helices $\alpha 3$ and $\alpha 4$ (aa 155 –

156), albeit weaker than the surrounding protein density presumably due to incomplete binding (**Suppl. Fig. 1B**). Closer inspection of the binding pocket and comparison with the PspA apo model revealed that the nucleotide competes with the $\alpha 3/\alpha 4$ loop, which results in an alternative conformation and reduced mobility of the $\alpha 3/\alpha 4$ loop, associated with lower atomic temperature factors, when the nucleotide is present (**Suppl. Fig. 1C**)."

Addition 2: We further investigated the critical role of the $\alpha 3/\alpha 4$ loop for nucleotide binding in the revised version of the manuscript by a deletion mutant that shows no ATPase activity and reduced plasticity (Refer to this Reviewer #1, point #3b, Addition 3 for details).

Addition 3: In the light of the Reviewer's comment and the observed incomplete binding and low ATPase activity, we now additionally discuss in the Discussion section of the manuscript:

"The low activity of PspA from *Synechocystis* is in accordance with the reported GTPase/ATPase activity of the closely related Vipp1/IM30 (Gupta et al., 2021; Junglas et al., 2020b; Ohnishi et al., 2018; Siebenaller et al., 2021). In support, low competition rates with well-characterized non-hydrolyzing ATP analogs reveal differences in ATP turnover when compared with canonical P-loop or Walker A motif-ATPases. However, when only few layers per rod molecule are considered actively hydrolyzing NTP, the effective turnover can be much higher locally. For example, considering the average length of PspA rods are $2.6 \pm 1.3 \mu\text{m}$, a total of approx. 10,000 monomers are found in rod corresponding to 1000 turns depending on the symmetry. Therefore, we reason that local ATP turnover can be approx. 1000 times higher, *i.e.*, three orders of magnitude, resulting apparent 50 ATP molecules per min (3000 h^{-1}), which is in the determined activity range of actively remodeling dynamin (Doo Song et al., 2004)."

Addition 4: The function of the ATPase activity with respect to membrane remodeling is included. (Refer to this Reviewer #1, point #3b, Addition 1 for details).

3a. The most compelling evidence is the fact that they observed other filament architectures in the presence of nucleotide; however, the process of grid freezing for cryo-EM is not highly reproducible, and these could merely be caused by slight changes in protein concentration or buffer composition.

We share the raised reproducibility concern of the Reviewer. As it is not common to determine exact replicates of structures in cryo-EM, we acquired five additional datasets of PspA for structural analysis in related conditions (added and summarized in Suppl. Table 2): one with ATP addition to preformed rods, two upon addition of non-hydrolyzable ATP analogs AMPPCP and ATP γ S, respectively, and two loop deletion mutants in the presence and absence of ATP. First, ATP addition to preformed rods showed a distinct distribution in comparison with assembling rods, which was actually suggested by Reviewer #2's point #2. In contrast, the other four data sets reveal close resemblance of diameter distributions for AMPPCP with previously recorded ADP (Suppl. Fig. 6C) and ATP γ S with previously recorded ATP (Suppl. Fig. 6E). Furthermore, the closest overlap in diameters was achieved by comparing two loop deletion mutant samples that are devoid of any ATP activity in the presence and absence of ATP (Fig. 5C). We believe that these independent comparisons now show that the process of grid freezing is indeed reliable and that the presented differences are significant.

3b. My concern here is that the authors are reporting non-specific binding of nucleotide, because of a positively-charged pocket, but that this is not related to its function. In the absence of functional data indicating how ATP binding and hydrolysis permits PspA function, this aspect of the manuscript is not entirely compelling.

Addition 1: In order to address the comment on PspA function, we linked the ATPase activity of PspA with functional data, by analyzing the PspA-induced membrane vesicle morphology changes by cryo-EM in the absence and presence of ATP (page 15, new Fig. 5, Suppl. Fig. 5):

“To assess the consequences of PspA’s structural plasticity in the context of a membrane remodeling activity, we analyzed the cryo-images for structural and morphological changes of the added EPL SUVs and the effect of ATP under reconstitution conditions developed for optimal tubulation (50 nm SUVs and *in situ* refolding) (**Fig. 5A and B**). In the absence of PspA, control SUVs had a narrow bilayer thickness distribution from 27 to 32 Å with a mean bilayer thickness of 31 Å (**Fig. 5C**). After incubation with PspA, we observed PspA rods engulfing and tubulating membrane. Notably, PspA rods engulfing membranes did not have uniform diameters, but were often attached to vesicles with one wider end and tapering towards the distal end (**Fig. 5B**). After incubation with PspA, SUV membranes were significantly thicker on average, with a broad distribution and a major peak at 38 Å and a minor peak at 68 Å. In the presence of PspA and ATP, the bilayer thickness distribution closely resembles the PspA sample (mean thickness 37 Å). The observed increase in bilayer thickness after incubation with PspA is consistent with a previous report (Junglas et al., 2021). To monitor changes in vesicle sizes, we used the vesicle perimeter as a descriptor (**Fig. 5D, Suppl. Fig. 5A**). The EPL SUVs alone had a mean perimeter of 100 nm corresponding to a mean diameter of 32 nm. Similar to the bilayer thickness, the distribution of vesicle perimeters was significantly increased in the presence of PspA and PspA+ATP, peaking at 160 nm and 150 nm, respectively. The vesicle size increase after incubation with PspA agrees with the previously reported PspA-mediated vesicle fusion (Junglas et al., 2021). Another prominent feature of the vesicles containing PspA was the formation of small, double-membrane vesicles (*i.e.*, small vesicles that are encapsulated by a larger vesicle), in analogous topology to intra-luminal vesicles (ILVs) (**Fig. 5B**, red arrowheads). Approx. 9 % of all vesicles were double-membrane vesicles in the control whereas the share of double-membrane vesicles was increased in the PspA preparations to 41 % and was highest in the PspA+ATP sample with 52 % (**Fig. 5E**). In addition, the distance between the two enclosed vesicle membrane, termed here the enclosure distance, was on average higher in the control vesicles (>70 Å) with a very broad enclosure distance distribution (**Suppl. Fig. 5B and C**). Both, the PspA as well as the PspA+ATP sample, showed a well-defined distance distribution with a mean enclosure distance of 67 and 62 Å, respectively. Compared with the vesicle perimeter, only the smallest vesicles (perimeter 100 – 200 nm) showed this constant enclosure distance of ~50 – 70 Å (**Suppl. Fig. 5D**). Together, quantitative image analysis of vesicle characteristics showed that the presence of ATP increases the occurrence of double-membrane vesicles by enhanced membrane remodeling capabilities of PspA.”

Addition 2: Please also refer to the response of Reviewer #2, point 2b where we characterized PspA with respect to binding of AMPCP and ATP γ S.

Addition 3: We designed a deletion mutant that is devoid of ATPase activity and shows reduced structural plasticity of diameter distributions (page 15, Fig. 6, Suppl. Fig. 6):

“In the series of determined PspA structures we found hinge 2, *i.e.*, the loop connecting helices α 3 and α 4 (loop α 3/ α 4), to be critically important for the conformational adjustments of PspA monomers in rods with higher diameters and also participating in the putative nucleotide binding region. Therefore, we generated a PspA mutant lacking ten residues of the loop E156 – S165 (PspA dL10) (**Fig. 6A**) and analyzed the ATPase activity as well as diameter distribution. First, PspA dL10 showed an ATPase activity reduced by over 90% compared with

wild-type PspA, although the above mutated residues critical for the ATPase activity (R44, E126, E179) were still intact (**Fig. 6B**). Second, we employed cryo-EM to determine the structure of PspA dL10 and compared it with the wild-type protein. We found that PspA dL10 still formed rods with symmetries and shapes also present in the wild-type sample, although with a narrower distribution centered at slightly higher diameters (**Fig. 6C and D**). Upon addition of ATP during the formation of PspA dL10 rods, we did not observe significant differences in the diameter distribution (220 ± 20 Å (dL10) vs. 230 ± 30 Å (dL10 ATP), in agreement with the highly reduced ATPase activity (**Fig. 6C and E, Suppl. Fig. G and H**). The detailed cryo-EM structures of PspA dL10 rods were almost identical to the wild-type rods, except for the missing loop densities between helices $\alpha 3$ and $\alpha 4$ (**Fig. 6F**). Together, the dL10 rod structures indicate that loop $\alpha 3/\alpha 4$ is not essential for the formation of PspA rods *per se*, but critical for the formation of rods with very low and high diameters, as well as for the associated ATPase activity. By removing the loop $\alpha 3/\alpha 4$, we reduced the conformational freedom providing an efficient means to restrict the observed structural plasticity of PspA assemblies.”

4. While I am supportive of publishing this work in NSMB, I would recommend to revise the manuscript, with much less emphasis on the ATPase binding and hydrolysis, and a much more cautious tone about its interpretation.

We thank the referee for supporting this manuscript for publication at NSMB. We hope that the included additional material also addresses the concern on the ATPase activity.

Reviewer #2 (Remarks to the Author):

In this manuscript, Junglas et al. investigate the molecular determinants of conformational flexibility in the ESCRT-III-family bacterial protein PspA. They use cryoEM to build atomic models for PspA filaments with various diameters and examine how the PspA monomer conformation and inter/intra-molecular interactions change to allow for the assembly of filaments of different diameters. This analysis reveals critical flexibility at hinge regions between alpha helices and specific pairwise interactions that either remain the same or change interacting residues in response to filament diameter changes. Finally, the authors show that, in vitro, PspA filaments display a low level of ATPase activity that may be related to changes in filament diameter. This work complements a body of literature showing that eukaryotic ESCRT-III proteins can assemble into various polymers and copolymers with highly variable curvature. These assemblies and their dynamics are thought to be mechanistically related to the roles of ESCRT proteins in reshaping membranes and catalyzing membrane fission. This manuscript is clearly written and contains numerous cryoEM structures and atomic models.

Major Issues:

1) The authors identify the hinge between $\alpha 3$ and $\alpha 4$ as moving the most between filaments of different diameters, followed by the hinge between $\alpha 2$ and $\alpha 3$. This result is consistent with the movement of the homologous hinge regions for ESCRT-III protein CHMP1B. Nguyen et al. NSMB 2020 (<https://doi.org/10.1038/s41594-020-0404-x>) showed that the very same hinges flex when comparing 180 and 280 Å diameter CHMP1B filaments. This prior work is not mentioned in the current manuscript. It is helpful to see that still further flexing of these same hinges accounts for the formation of still wider PspA filaments and, therefore, that this mechanism is conserved across domains of life in the ESCRT-III family. However, Junglas et al. should frame their results within the context of the published work that established this conceptual precedent.

We apologize for the oversight in not referencing this similarity between PspA's plasticity and CHMP1B structural flexibility. This study even adds indirect support for our observation. We follow the suggestion, mention the conformational states, thereby frame it around the Nguyen work while also introducing the differences to our observed plasticity. We now added this study to the introduction and included a paragraph on this topic in the discussion:

- Introduction (page 4):
“Eukaryotic ESCRT-III monomers have been observed in different conformational states in formed heteropolymeric complexes of different ESCRT-III isoforms (Azad et al., 2023; Nguyen et al., 2020). However, the structural flexibility of ESCRT-III proteins in homopolymers and the enabling structural mechanisms have not been studied in molecular detail.”
- Discussion (page 20):
“In fact, the structural changes observed now for the hinges connecting helices α_3 and α_4 as well as helices α_2 and α_3 are consistent with structures of 180 Å CHMP1B/IST1 copolymer and 280 Å diameter CHMP1B filaments showing IST1 binding-induced conformational changes in the homologous hinge regions of the ESCRT-III protein CHMP1B (Nguyen et al., 2020).”

Together, the data presented in our manuscript including the revisions go significantly beyond the structural observations of Nguyen et al. 2020.

2) The observation that PspA can hydrolyze ATP in vitro is interesting, but more should be done to characterize the role, if any, this activity has in filament structure.

2a. For example, ADP/ATP were always added during PspA refolding and assembly. What is the effect on filament structure if the ATP is added after filament assembly?

In order to address this comment, we recorded a cryo-EM dataset of preformed PspA rods incubated with ATP.

On page 15, we added to the manuscript:

“To further investigate how nucleotide binding and hydrolysis affects the PspA rod diameter distribution, we next analyzed the rod diameter distribution after addition of ATP to preformed PspA rods using cryo-EM. The resulting rod distribution showed two distinct peaks at 215 Å and 250 Å diameters whereas only a minor shift to larger diameters, compared with the rod distribution after the addition of ATP during rod assembly, was observed (**Suppl. Fig. 6A and B**). These observations indicate that only minor diameter changes are possible once the rods have formed and larger changes can only occur during rod assembly or re-assembly.”

2b. Does adding AMPPCP or ATP γ S during filament formation in the presence of ATP shift the distribution of filament diameters towards smaller diameters?

In the previous version of the manuscript, we performed the proposed experiments to measure the associated ATPase activity with increasing concentrations of AMPPCP and ATP γ S (Suppl. Fig. 4C). Upon increasing AMPPCP concentrations, the determined ATPase activity remains around 2.0 h⁻¹ while ATP γ S was capable of inhibiting PspA's ATPase activity, albeit at high concentrations. These experiments suggest that there is only small competition between ATP and AMPPCP/ATP γ S and associated changes in diameter populations may be challenging to detect. Therefore, we incubated PspA directly with AMPPCP as well as ATP γ S and included those data in the manuscript on page 15 as well as in the new Suppl. Fig 6C-5F:

“Additionally, we tested the effect of the non-hydrolyzable ATP analogs AMPPCP and ATP γ S on the diameter distribution using cryo-EM. First, upon addition of AMPPCP during rod formation, we found that the diameter distribution is approximately indistinguishable from the distribution in the presence of ADP supported by the average \pm standard deviation measurements: 240 ± 40 Å and 240 ± 50 Å of the ADP and the AMPPCP distribution, respectively (**Suppl. Fig. 6C and D**). Unlike for canonical ATPases (Krasteva and Barth, 2007), these data indicate that AMPPCP mimics the ADP-bound state in the case of PspA. Second, upon addition of ATP γ S, the diameter distribution overlapped highly with the ATP sample although that showed an additional tail of higher diameter rods (>300 Å) (**Suppl. Fig. 6E and F**). Assuming that ATP γ S mimics the ATP transition state, this observation suggests that complete ATP turnover rather than mere binding is required for the formation of rods with large diameters.”

Minor Issues:

1) Line 89, “cyanobacterium” is misspelled.

Corrected.

2) In discussing ATP usage by membrane remodeling proteins, the authors may want to consider discussing the case of the dynamin family protein EHD1. Unlike other dynamin family proteins and in analogy to PspA, ATP hydrolysis causes an expansion of the helical filaments (Deo et al. Nat Comm 2018, <https://doi.org/10.1038/s41467-018-07586-z>).

We thank this Reviewer mentioning this reference that we now included in our manuscript. We added a respective paragraph to our discussion.

Decision Letter, first revision:**Message:** 15th Jan 2024

Dear Dr. Sachse,

Thank you again for submitting your manuscript "Structural plasticity of bacterial ESCRT-III protein PspA in higher order assemblies". I apologize for the delay in responding, which resulted from the difficulty in obtaining suitable referee reports. Nevertheless, we now have comments (below) from the 2 reviewers who evaluated your paper. In light of those reports, we remain interested in your study and would like to see your response to the comments of the referees, in the form of a revised manuscript.

You will see that while the reviewers appreciate the revised manuscript, they have some remaining concerns. We would ask you for one final effort in revising the paper. Specifically, we agree with reviewer #1 that inclusion of an additional control will further strength the manuscript, if feasible. The experiment could also in part assuage concerns of reviewer #2. While we agree with reviewer #2 that extending the work to in vivo systems would solidify the conclusions, we do not find it necessary in the context of the current work.

Please be sure to address/respond to all concerns of the referees in full in a point-by-point response and highlight all changes in the revised manuscript text file. If you have comments that are intended for editors only, please include those in a separate cover letter.

We expect to see your revised manuscript within 8 weeks. If you cannot send it within this time, please contact us to discuss an extension; we would still consider your revision, provided that no similar work has been accepted for publication at NSMB or published elsewhere.

Reporting Summary:

When submitting the revised version of your manuscript, please pay close attention to our [href="https://www.nature.com/nature-portfolio/editorial-policies/image-integrity">Digital Image Integrity Guidelines](https://www.nature.com/nature-portfolio/editorial-policies/image-integrity). and to the following points below:

Please note that all key data shown in the main figures as cropped gels or blots should be presented in uncropped form, with molecular weight markers. These data can be aggregated into a single supplementary figure item. While these data can be displayed in a relatively informal style, they must refer back to the relevant figures. These data should be submitted with the final revision, as source data, prior to acceptance, but you may want to start putting it together at this point.

Data availability: this journal strongly supports public availability of data. All data used in accepted papers should be available via a public data repository, or alternatively, as Supplementary Information. If data can only be shared on request, please explain why in your Data Availability Statement, and also in the correspondence with your editor. Please note that for some data types, deposition in a public repository is mandatory - more information on our data deposition policies and available repositories can be found below: <https://www.nature.com/nature-research/editorial-policies/reporting-standards#availability-of-data>

We require deposition of coordinates (and, in the case of crystal structures, structure factors) into the Protein Data Bank with the designation of immediate release upon publication (HPUB). Electron microscopy-derived density maps and coordinate data must be deposited in EMDB and released upon publication. Deposition and immediate release of NMR chemical shift assignments are highly encouraged. Deposition of deep sequencing

and microarray data is mandatory, and the datasets must be released prior to or upon publication. To avoid delays in publication, dataset accession numbers must be supplied with the final accepted manuscript and appropriate release dates must be indicated at the galley proof stage.

Nature Structural & Molecular Biology is committed to improving transparency in authorship. As part of our efforts in this direction, we are now requesting that all authors identified as 'corresponding author' on published papers create and link their Open Researcher and Contributor Identifier (ORCID) with their account on the Manuscript Tracking System (MTS), prior to acceptance. This applies to primary research papers only. ORCID helps the scientific community achieve unambiguous attribution of all scholarly contributions. You can create and link your ORCID from the home page of the MTS by clicking on 'Modify my Springer Nature account'. For more information please visit please visit www.springernature.com/orcid.

[Redacted]

Sincerely,
Kat

Katarzyna Ciazynska, PhD
(she/her)
Associate Editor
Nature Structural & Molecular Biology
<https://orcid.org/0000-0002-9899-2428>

Reviewers' Comments:

Reviewer #1:

Remarks to the Author:

In this revised manuscript, Jungas and co-workers provide two major new elements to their study on the membrane-remodelling protein PspA:

- New analysis of their cryo-EM maps, as well as additional structures with mutants, provide direct evidence of the presence of DNA bound to PspA. This renders that part of the manuscript a lot more convincing.

- In addition, they use a membrane fusion assay to demonstrate that ATP binding alters the propensity of the PspA filaments to induce the formation of double-membrane vesicles, thus demonstrating the functional relevance of DNA binding. Collectively, this forms a very impressive manuscript, very well suited for publication in NSMB.

I do have a couple of minor suggestions, to help readers follow the story:

- I actually don't find figure 6F very clear; one could interpret it into thinking that the density shown with red arrows correspond to the L10 loop density. I find the data shown in Supplementary figure 1C a lot more convincing that there is in fact density for ATP. I recommend switching these two panels.
- In addition, it would be very helpful to have a close-up view of the nucleotide, and its coordination, somewhere in the manuscript. This would help identify the roles of E179, E126 and R44 in coordination, and to see if there are residues interaction with the Adenine. Is there density for a magnesium ion?
- On the vesicle morphology assay, I'm very curious as to what the mechanism for how PspA induces double-membrane formation might be? While this is probably largely speculative, a paragraph about this in the discussion might be of interest.
- I would suggest to add an additional control for this experiment: a cryo-EM dataset of vesicles formed in the presence of SynPspAR44K/E126Q/E179Q + ATP. This would really help convince that the effect observed is due to the difference in PspA structure/biophysical properties, and not to ATP present in the buffer. I do understand that this is a lot of work, and I don't think it is critical, but would be a neat control to add, if possible.

Julien Bergeron

Reviewer #2:

Remarks to the Author:

In their revised manuscript, Junglas et al. add a large amount of new data and discussion related to the ATPase activity of PspA, and the impact of the nucleotide on PspA assemblies. The evidence for ATP binding and hydrolysis by PspA filaments is now more robust. This includes a revised analysis of the putative ATP density in some cryoEM maps and the dL10 mutant, which ablates ATPase activity despite leaving catalytic residues intact. The dL10 mutant is further interesting because it restricts the conformational plasticity of the PspA rods. The observed effects of nonhydrolyzable ATP analogs are also consistent with a role for ATP hydrolysis in modulating the diameter of PspA rods. This part and the original description of molecular changes underlying PspA rod size plasticity are convincing aspects of the revised manuscript.

Next, the authors attempt to link ATP binding and hydrolysis to PspA function by showing that the presence of ATP alters PspA's membrane remodeling activity. PspA appears to change the morphology of in vitro vesicles, and some of these effects are modestly altered by adding ATP. We find this addition to the manuscript less convincing because the increase in the percentage of double-membrane vesicles in the PspA + ATP sample versus PspA alone sample is modest and remain mechanistically obscure. Moreover, there is no explanatory link between the in vitro activity consequences of PspA-induced changes in apparent liposome morphology and PspA function(s) in cells. We realize that testing function inside of Synechocystis is difficult and would be beyond the scope of the current

study.

Specific points:

In Fig. 5B, it is not clear to us where the engulfed vesicles are. Are they in the form of membrane nanotubes within the PspA rods, or are they what appear to be pieces of lipid bilayer within some of the filaments? It would be helpful to explicitly identify the fate(s) of lipids engulfed by PspA rods in this figure with arrows or something similar.

Author Rebuttal, first revision:

We thank the referees for the positive and constructive feedback on our manuscript NSMB-A47957. We made an extra effort to complete the raised concerns. We amended the manuscript text and figures when requested. Most notably, we added experimental data of an ATPase-deficient mutant as a control of the vesicle morphology assay to consolidate the claim that PspA's ATP binding/hydrolysis enhances the membrane remodeling activity.

Reviewers' Comments:

Reviewer #1:

Remarks to the Author:

In this revised manuscript, Jungas and co-workers provide two major new elements to their study on the membrane-remodelling protein PspA:

- New analysis of their cryo-EM maps, as well as additional structures with mutants, provide direct evidence of the presence of DNA bound to PspA. This renders that part of the manuscript a lot more convincing.
- In addition, they use a membrane fusion assay to demonstrate that ATP binding alters the propensity of the PspA filaments to induce the formation of double-membrane vesicles, thus demonstrating the functional relevance of DNA binding

Collectively, this forms a very impressive manuscript, very well suited for publication in NSMB.

I do have a couple of minor suggestions, to help readers follow the story:

1. I actually don't find figure 6F very clear; one could interpret it into thinking that the density shown with red arrows correspond to the L10 loop density. I find the data shown in Supplementary figure 1C a lot more convincing that there is in fact density for ATP. I recommend switching these two panels.

We believe that this point arises from a misunderstanding. Figure 6F is intended to highlight the presence of L10 loop density in the WT map in comparison with the dL10 map. It is not supposed to show the presence of nucleotide density, which is shown in Supp Fig. 1C. The Figure 6F legend also indicates that the red arrowheads point to the loop density: "Comparison of the cryo-EM density maps of PspA+ATP (grey) and PspA dL10 (green). Red arrowheads indicate wildtype density for the loop that is missing in the dL10 map". We also clarify in the text on page 14: "The detailed cryo-EM structures of PspA dL10 rods were almost identical to the wild-type rods, while the density for the loop between helices $\alpha 3$ and $\alpha 4$ was missing (Fig. 6F)."

2. In addition, it would be very helpful to have a close-up view of the nucleotide, and its coordination, somewhere in the manuscript. This would help identify the roles of E179, E126 and R44 in coordination, and to see if there are residues interaction with the Adenine. Is there density for a magnesium ion?

Suppl. Fig. 1 B/C show a series of close-up views of the nucleotide and its coordination. The relevant residues R44, E126 and E179 are shown and labeled in Suppl. Fig. 1B (right panel). As we could not identify density for Mg^{2+} we refrained from including it in the model/coordination figures. Also please note, that PspA is also active in absence of Mg^{2+} (Suppl. Fig. 4B), thus it is not entirely clear whether Mg^{2+} is involved in coordination of the nucleotide.

3. On the vesicle morphology assay, I'm very curious as to what the mechanism for how PspA induces double-

membrane formation might be? While this is probably largely speculative, a paragraph about this in the discussion might be of interest.

As described in detail in our previous work (<https://doi.org/10.1016/j.cell.2021.05.042>), we suggest an intraluminal budding process for the formation of the double-membrane vesicles. We now briefly mention the mechanism of this process and refer to our previous model for a detailed description:

PspA rods mixed with liposomes produced double-membrane vesicles that share the same topology as ILVs and are likely a result of inward-vesicle budding, i.e., an intraluminal budding process (Junglas et al., 2021). In accordance with our previous work, we suggest that double-membrane vesicles are formed by cross-membrane vesicle transfer, i.e., by releasing of PspA-engulfed membrane into an acceptor vesicle. In the current work, we demonstrated that after PspA incubation the number of double-membrane vesicles increased in the presence of ATP. This observation indicates that the membrane remodeling activity of PspA is stimulated by ATP, i.e., by facilitating the uptake of vesicle membranes when larger diameter rods engulf membranes more efficiently than low diameter rods due to lowered membrane curvature.

4. I would suggest to add an additional control for this experiment: a cryo-EM dataset of vesicles formed in the presence of SynPspAR44K/E126Q/E179Q + ATP. This would really help convince that the effect observed is due to the difference in PspA structure/biophysical properties, and not to ATP present in the buffer. I do understand that this is a lot of work, and I don't think it is critical, but would be a neat control to add, if possible.

As suggested by this reviewer, we added an additional control experiment with the inactive PspA R44K/E126Q/E179Q mutant in the presence of ATP and EPL SUVs. In this experiment, the number of formed double-membrane vesicles was significantly lower than the number of double-membrane vesicles in the sample with WT PspA (30% vs 52%). Therefore, we now added the new data to Fig. 5B and the following paragraph to the manuscript:

Using the ATPase-deficient mutant PspA R44K/E126Q/E179Q as a negative control, we validated that indeed ATPase activity of PspA and not the addition of ATP gave rise to the increased number of double-vesicles in the PspA+ATP sample.

Reviewer #2:

Remarks to the Author:

In their revised manuscript, Junglas et al. add a large amount of new data and discussion related to the ATPase activity of PspA, and the impact of the nucleotide on PspA assemblies. The evidence for ATP binding and hydrolysis by PspA filaments is now more robust. This includes a revised analysis of the putative ATP density in some cryoEM maps and the dL10 mutant, which ablates ATPase activity despite leaving catalytic residues intact. The dL10 mutant is further interesting because it restricts the conformational plasticity of the PspA rods. The observed effects of nonhydrolyzable ATP analogs are also consistent with a role for ATP hydrolysis in modulating the diameter of PspA rods. This part and the original description of molecular changes underlying PspA rod size plasticity are convincing aspects of the revised manuscript.

1. Next, the authors attempt to link ATP binding and hydrolysis to PspA function by showing that the presence of ATP alters PspA's membrane remodeling activity. PspA appears to change the morphology of in vitro vesicles, and some of these effects are modestly altered by adding ATP. We find this addition to the manuscript less convincing because the increase in the percentage of double-membrane vesicles in the PspA + ATP sample versus PspA alone sample is modest and remain mechanistically obscure. Moreover, there is no explanatory link between the in vitro activity consequences of PspA-induced changes in apparent liposome morphology and PspA function(s) in cells. We realize that testing function inside of *Synechocystis* is difficult and would be beyond the scope of the current study.

As suggested by reviewer #1 point 4, we added an additional control experiment with the inactive mutant PspA R44K/E126Q/E179Q to further strengthen the observation that PspA's ATP binding/hydrolysis enhances the membrane remodeling activity of PspA. We refrained from speculating about the details of the link between liposome morphology changes and in vivo activity of PspA, because data on that issue is scarce. We agree with this reviewer that elucidating the in vivo function and effect of PspA on membranes in more detail would be highly interesting, but also is beyond the scope of the current study.

Specific points:

2. In Fig. 5B, it is not clear to us where the engulfed vesicles are. Are they in the form of membrane nanotubes within the PspA rods, or are they what appear to be pieces of lipid bilayer within some of the filaments? It would be helpful to explicitly identify the fate(s) of lipids engulfed by PspA rods in this figure with arrows or something similar.

We agree with this reviewer that engulfed membranes in the provided micrograph are difficult to spot. As suggested, we added additional arrows for engulfed membranes in Figure 5B and added the following paragraph to the manuscript to further describe the fates of engulfed membranes:

Notably, PspA rods engulfing membranes did not have uniform diameters. Instead, they were frequently attached to vesicles with one wider end tapering towards the distal end (Fig. 5B). Moreover, engulfed membranes were continuous lipid tubes as well as small, isolated vesicles and membrane patches.

Decision Letter, second revision:

Message: Our ref: NSMB-A47957B

25th Mar 2024

Dear Dr. Sachse,

Thank you for submitting your revised manuscript "Structural plasticity of bacterial ESCRT-III protein PspA in higher order assemblies" (NSMB-A47957B). It has now been seen by the original referees and their comments are below. The reviewers find that the paper has improved in revision, and therefore we'll be happy in principle to publish it in Nature Structural & Molecular Biology, pending minor revisions to satisfy the referees' final requests and to comply with our editorial and formatting guidelines.

Sincerely,

Katarzyna Ciazynska, PhD
(she/her)
Associate Editor
Nature Structural & Molecular Biology
<https://orcid.org/0000-0002-9899-2428>

Reviewer #1 (Remarks to the Author):

The Authors have now very extensively addressed all the points raised by the reviewers,

and I am delighted to recommend this manuscript for publication in NSMB.

Reviewer #2 (Remarks to the Author):

The authors have addressed my questions and concerns.

Final Decision Letter: